# Generation of DNA oligomers with similar chemical kinetics via in-silico optimization

Michael Tobiason [1,2✉], Bernard Yurke [2,3] & William L. Hughes [4✉]

Networks of interacting DNA oligomers are useful for applications such as biomarker detection, targeted drug delivery, information storage, and photonic information processing. However, differences in the chemical kinetics of hybridization reactions, referred to as kinetic dispersion, can be problematic for some applications. Here, it is found that limiting unnecessary stretches of Watson-Crick base pairing, referred to as unnecessary duplexes, can yield exceptionally low kinetic dispersions. Hybridization kinetics can be affected by unnecessary intra-oligomer duplexes containing only 2 base-pairs, and such duplexes explain up to 94% of previously reported kinetic dispersion. As a general design rule, it is recommended that unnecessary intra-oligomer duplexes larger than 2 base-pairs and unnecessary inter-oligomer duplexes larger than 7 base-pairs be avoided. Unnecessary duplexes typically scale exponentially with network size, and nearly all networks contain unnecessary duplexes substantial enough to affect hybridization kinetics. A new method for generating networks which utilizes in-silico optimization to mitigate unnecessary duplexes is proposed and demonstrated to reduce in-vitro kinetic dispersions as much as 96%. The limitations of the new design rule and generation method are evaluated in-silico by creating new oligomers for several designs, including three previously programmed reactions and one previously engineered structure.

[1] Department of Computer Science, Boise State University, Boise, ID, USA. [2] Micron School of Materials Science & Engineering, Boise State University, Boise, ID, USA. [3] Department of Electrical & Computer Engineering, Boise State University, Boise, ID, USA. [4] School of Engineering, University of British Columbia Okanagan Campus, Kelowna, BC, Canada. ✉email: michaeltobiason@u.boisestate.edu; will.hughes@ubc.ca

Molecules of single stranded deoxyribonucleic acid, sometimes referred to as DNA oligomers, can form duplexes where two complementary base-sequences are held together by Watson-Crick base-pairing. Networks of DNA oligomers interacting through intentional duplexes have been used to fabricate structures[1–7] and program chemical reactions[8,9]. Demonstrated applications for such networks include biomarker detection[10,11], genomic sequencing[12], targeted drug delivery[13–15], artificial gene regulation[16,17], information storage[18,19], and photonic information processing[20,21]. Hybridization reactions (i.e., reactions where DNA oligomers form new duplexes) are fundamental to these networks. However, DNA oligomers undergoing hybridization reactions sometimes exhibit a large kinetic dispersion (i.e., a large difference in chemical kinetics)[12,22,23]. Large kinetic dispersions may cause issues such as false negatives during biomarker detection[10,11], limited throughput of genomic sequencing[12], failure to release medication[13–15], under-expression of a regulated gene[16,17], or loss of stored information[18,19]. Large kinetic dispersions may also contribute to the inconsistent structure formation[3,24–26] and high development costs[4,27–30] sometimes associated with networks of DNA oligomers.

A given network of DNA oligomers is often intended to form specific duplexes which either implement a chemical reaction network[31,32] or create a spatial arrangement of matter (i.e., a two-dimensional[7,33,34] or three-dimensional[35] shape). Such networks may also form other duplexes, referred to here as unnecessary duplexes. It has been previously established that unnecessary duplexes can affect hybridization reactions[9,23,36–40], making them one potential contributor to kinetic dispersion. However, several other factors known to affect hybridization reactions may also contribute to kinetic dispersion. For separate aqueous solutions which contain oligomers with the same base-sequences, initial state, and final state, these known factors include temperature[12,22,41–44], ionic strength[45,46], and viscosity[45,47]. For separate aqueous solutions containing oligomers with different base-sequences but similar initial and final states, these known factors additionally include oligomer length[42] and duplex stability[12,22,48,49].

Here, the relationship between unnecessary duplexes and kinetic dispersion is studied using a combination of in-silico and in-vitro methods. First, by analyzing existing experimental data, it is found that unnecessary intra-oligomer duplexes are a key contributor to kinetic dispersion and that nearly all previously reported kinetic dispersion can be explained by known causes. It is recommended that when generating new networks all unnecessary intra-oligomer duplexes larger than 2 base-pairs and all unnecessary inter-oligomer duplexes larger than 7 base-pairs be avoided if possible. Satisfying both of these conditions is referred to as the "no 3's and no 8's" design rule. By randomly sampling networks in-silico, it is found that nearly all networks contain unnecessary intra-oligomer duplexes substantial enough to cause kinetic dispersion. A new network generation method which utilizes in-silico optimization to mitigate unnecessary duplexes is proposed and is demonstrated to successfully reduce in-vitro kinetic dispersion by as much as 96%. Finally, the limitations of both the "no 3's and no 8's" design rule and the generation method are studied in-silico by generating new networks for several designs. It is found that the new generation method substantially increases the designs for which a researcher can satisfy the new design rule, presumably enabling reduced kinetic dispersions and more reliable performance for future applications.

## Results

### Analysis of kinetic dispersion in existing experimental data.
Numerous previous studies have characterized the chemical kinetics of DNA oligomers undergoing a hybridization reaction, and some of these studies characterized enough samples for a meaningful re-analysis of the data[12,22,23]. Here, this existing experimental data was used to ascertain how much kinetic dispersion can be explained by unnecessary duplexes and to better understand the relationship between unnecessary duplexes and kinetic dispersion.

Five datasets (i.e., sets of comparable rate-constant values) were identified for analysis and are listed in Fig. 1a. The hybridization reactions within these datasets range from a relatively simple duplex-formation reaction to a relatively complicated multiple-intermediate catalytic reaction. Other than unnecessary duplexes, all known hybridization-reaction-affecting factors (including temperature, ionic strength, viscosity, oligomer length, and duplex-stability) were approximately constant within each dataset. Each dataset was given a label abbreviating the first author, temperature, and reaction type. For example, the values measured by Zhang et al.[12] at 37 °C for a duplex-formation reaction were labeled Z37F.

Histograms of the rate-constant values in each dataset are reported within Fig. 1a. Each rate-constant distribution is left skewed with the mean less than the median and appears poorly approximated by either a Gaussian distribution or a Galton distribution. Each rate-constant distribution spans at least 3 orders of magnitude, verifying the presence of substantial kinetic dispersion. The kinetic dispersion of each dataset was further quantified using the Inter-Quartile Range of the Natural Logarithm of the rate-constant values (denoted IQRNL and detailed in the methods section). Supplementary note 1 reports additional information from the analysis of these datasets, with IQRNL values for each dataset reported in supplementary table S1. The rate-constant values underlying each histogram can be found in the 'Supplementary Data 1' file.

Sub-populations with the least substantial unnecessary intra-oligomer duplexes were identified using the oligomer fitness score above baseline (denoted $\Delta O$ and detailed in the methods section). This fitness score penalizes a network with $10^L$ fitness points for each unnecessary intra-oligomer duplex of length $L$. A $\Delta O$ value of 0 indicates no unnecessary intra-oligomer duplexes, and increasing $\Delta O$ values indicate more substantial unnecessary duplexes. Plots of rate-constant value as a function of $\Delta O$ are reported in supplementary fig. S1. Detailed statistical analyses of the most $\Delta O$-fit sub-populations with number of samples (n) equal to 2, 3, 4, 8 and 16 are reported in supplementary figs. S2 to S6. The rate-constant values of the 3 most $\Delta O$-fit samples are highlighted in orange in Fig. 1a.

A different underlying rate-constant distribution was clearly resolved for many of the $\Delta O$-fit sub-populations (i.e., the sub-populations with the least substantial unnecessary intra-oligomer duplexes). The null hypothesis was established that a given sub-population was drawn from the same underlying distribution as the other samples in the dataset. When a Kolmogrov–Smirnov[50] test with a threshold of $p = 0.05$ was used to test this null hypothesis, it was successfully rejected for 21 of the 25 $\Delta O$-fit sub-populations. For the sub-populations containing the 3 most $\Delta O$-fit samples, this null hypothesis was successfully rejected for 4 of the 5 datasets. All of the $\Delta O$-fit sub-populations which failed to reject the null hypothesis were from the O25C dataset, which may be explained by the proximity of the peaks of these rate-constant distributions. When the null hypothesis was instead tested using a Wilcoxon-signed-rank test[51] or an Anderson-Darling test[52] similar trends were observed, however the Wilcoxon-signed rank test required larger sample sizes to achieve statistical significance.

Lower kinetic dispersions were clearly resolved for many of the $\Delta O$-fit sub-populations (i.e., the sub-populations with the least substantial unnecessary intra-oligomer duplexes). For the 3 most $\Delta O$-fit samples, kinetic dispersions were reduced 84%, 92%, 86%,

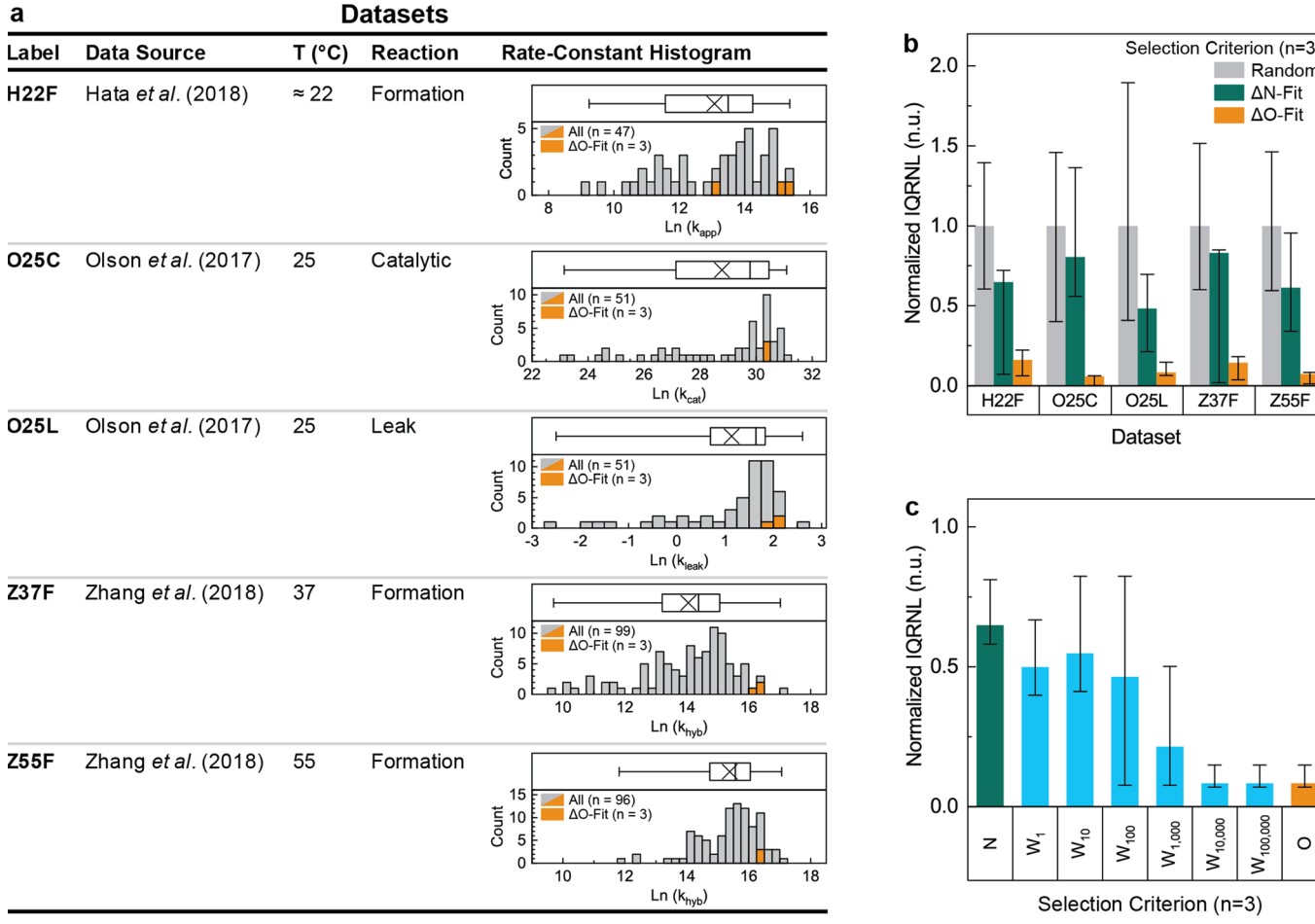

**Fig. 1 Analysis of existing in-vitro rate-constants. a** The five datasets identified for analysis. Histograms indicate the number of rate-constants in each 0.25 interval. Orange highlights the 3 most ΔO-fit samples. Box charts shown above each histogram indicate the mean (x within box), the 50th percentile (line within box), 25th / 75th percentiles (left end and right end of box), and min/max values (brackets extending from box). Kinetic dispersion was quantified using the interquartile range of the natural logarithm of rate-constants (IQRNL), which is visible as the distance between the left end and right end of the box. **b** Relative kinetic dispersions estimated for select sub-populations from each dataset. The lower error bar, the column height, and the upper error bar represent the 25th, 50th, and 75th percentiles of each estimate. **c** Median relative kinetic dispersions for select fitness scores. The lower error bar, the column height, and the upper error bar represent the 25th, 50th, and 75th percentiles.

93%, and 94% relative to equally sized random sub-populations. The value and bounds of these estimates are reported in supplementary table S2 and Fig. 1b (orange bars). No overlap existed between the bounds of these kinetic dispersion estimates and those of randomly selected sub-populations.

The large kinetic dispersion reductions for ΔO-fit sub-populations were interpreted as evidence that the kinetic dispersion in each dataset can be largely explained by unnecessary duplexes contributing to ΔO. Since any intra-oligomer duplex logically implies the existence of two inter-oligomer duplexes, this includes at least some inter-oligomer duplexes. These reductions also verify that the experimental methods of the prior researchers mitigated other causes of kinetic dispersion sufficiently that unnecessary intra-oligomer duplexes explain most of the remaining kinetic dispersion.

Sub-populations with the least substantial unnecessary inter-oligomer duplexes were identified using the network fitness score above baseline (denoted ΔN and detailed in the methods section). This fitness score penalizes a network with $10^L$ fitness points for each unnecessary inter-oligomer duplex of length $L$. A ΔN value of 0 indicates no unnecessary inter-oligomer duplexes, and increasing ΔN values indicate more substantial unnecessary duplexes. Plots of rate-constant value as a function of ΔN are

reported in supplementary fig. S7. Detailed statistical analyses of the most ΔN-fit sub-populations with a number of samples equal to 2, 3, 4, 8 and 16 are reported in supplementary figs. S8 to S12.

A different underlying rate-constant distribution was not clearly resolved for most ΔN-fit sub-populations (i.e., the sub-populations with the least substantial unnecessary inter-oligomer duplexes). The null hypothesis was established that a given sub-population was drawn from the same underlying distribution as the other samples in the dataset. When a Kolmogorov–Smirnov test[50] with a threshold of $p = 0.05$ was used to test this null hypothesis, it was successfully rejected for only 1 of the 25 ΔO-fit sub-populations. For the sub-populations containing the 3 most ΔN-fit samples, this null hypothesis was successfully rejected for 0 of the 5 datasets. When the null hypothesis was instead evaluated using a Wilcoxon-signed-rank test[51] or an Anderson–Darling test[52] similar trends were observed.

Only marginally reduced kinetic dispersions were observed for the ΔN-fit sub-populations (i.e., the sub-populations with the least substantial unnecessary inter-oligomer duplexes). For sub-populations containing the three most ΔN-fit samples ($n = 3$, Fig. 1b green bars), kinetic dispersions were reduced 37%, 51%, 23%, 39%, and 19% relative to equally sized random sub-populations (Fig. 1b grey bars). The value and bounds of these

estimates are reported in supplementary table S2 and Fig. 1b (green bars). For all 5 datasets, the bounds of these kinetic dispersion estimates overlapped with those of randomly selected sub-populations. The systematic decrease in kinetic dispersion across all five datasets was interpreted as evidence that at least some of the kinetic dispersion in these datasets may be explained by unnecessary duplexes contributing to $\Delta N$. This may be explained by the fact that each intra-oligomer duplex logically implies the existence of two inter-oligomer duplexes.

For both the $\Delta O$-fit and the $\Delta N$-fit sub-populations, smaller sample sizes generally correlated with larger kinetic dispersion reductions. This trend is consistent with attempting to select a small number of fit samples from a larger population. Since the kinetic dispersion reductions were calculated relative to equally sized random sub-populations, it is unlikely that this trend is caused by biased parameter estimation. While the minimum number of samples necessary to estimate IQRNL is 2 (i.e., $n = 2$), $n = 3$ often yielded lower kinetic dispersions. This may be explained by the different definitions of median for even and odd numbers of samples.

The unnecessary intra-oligomer duplexes present to explain the observed kinetic dispersion indicate that relatively small intra-oligomer duplexes are substantial enough to cause kinetic dispersion. For all 5 datasets, unnecessary intra-oligomer duplexes with 3 or more base-pairs exist to explain the increased kinetic-dispersion of non $\Delta O$-fit samples. However, the 16 most $\Delta O$-fit samples in the H22F dataset exhibited relatively large kinetic dispersions, yet these samples contain no unnecessary intra-oligomer duplex larger than 2 base-pairs. This was interpreted as evidence that unnecessary 3-base-pair intra-oligomer duplexes are substantial enough to affect hybridization kinetics under typical experimental conditions, and that unnecessary 2-base-pair intra-oligomer duplexes may be substantial enough to affect hybridization kinetics under certain experimental conditions.

The $\Delta O$ difference associated with these kinetic dispersion reductions indicates that relatively small $\Delta O$ values are necessary to limit kinetic dispersion. For the 5 datasets, the $\Delta O$ of the three most $\Delta O$-fit samples differed from other samples by a median of $3.3 \times 10^3$, $7.0 \times 10^3$, $7.0 \times 10^3$, $1.0 \times 10^5$, and $1.1 \times 10^5$ fitness points. The smallest of these differences ($3.3 \times 10^3$) came from the H22F dataset and was associated with an 84% reduction in kinetic dispersion, suggesting that a $\Delta O$ difference of $3.3 \times 10^3$ is sufficient to cause substantial kinetic dispersion. Since each unnecessary 3-base-pair intra-oligomer duplex contributes $1 \times 10^3$ fitness points to $\Delta O$, the value of $3.3 \times 10^3$ is equivalent to approximately 3 unnecessary 3-base-pair duplexes.

Unnecessary intra-oligomer duplexes may also explain the marginal kinetic dispersion reductions observed for the most $\Delta N$-fit sub-populations. The three most $\Delta N$-fit samples in each dataset exhibited kinetic dispersions marginally lower than random samples, and the $\Delta O$ of these $\Delta N$-fit samples differed from other samples by a median of $2.4 \times 10^3$, $4 \times 10^3$, $4 \times 10^3$, $4 \times 10^4$, and $5 \times 10^4$ fitness points. These values are large enough that the observed correlation between decreasing $\Delta N$ and decreasing kinetic dispersion may be explained by reduced unnecessary intra-oligomer duplexes. The decreased $\Delta O$ for $\Delta N$-fit samples can be explained by the fact that each intra-oligomer duplex logically implies the existence of two inter-oligomer duplexes.

Kinetic dispersions were also estimated for the most fit sub-populations according to several versions of the weighted fitness score above baseline (abbreviated $\Delta W_x$ and detailed in the methods section). These fitness scores are a weighted linear combination of $\Delta N$ and $\Delta O$ governed by the weighting parameter x and can be used to simultaneously quantify intra-oligomer and inter-oligomer unnecessary duplexes. A $\Delta W_x$ fitness score with a specific x value is denoted by incorporating the x value into the subscript (for example, $\Delta W_x$ with x = 10,000 is denoted as $\Delta W_{10,000}$). For all $\Delta W_x$, a value of 0 indicates no unnecessary duplexes, and increasing values indicate more substantial unnecessary duplexes. Kinetic dispersion estimates for the most $\Delta W_x$-fit sub-populations for a sample-size of $n = 3$ are reported in Fig. 1c. For $x \geq 10^4$ the most $\Delta W_x$-fit sub-populations were identical to the most $\Delta O$-fit sub-populations, and the same large kinetic dispersion reductions were observed. For $x \leq 10^2$, the most $\Delta W_x$-fit samples were similar or identical to the most $\Delta N$-fit samples, and similarly yielded only marginal kinetic dispersion reductions. These results were expected based on the definition of $\Delta W_x$ and demonstrate how larger x values can be used to increase the penalty for unnecessary intra-oligomer duplexes. Since $\Delta W_x$ with $x \geq 10^4$ performed sufficiently for all five datasets, it was inferred that this fitness score may be useful across a range of experimental conditions and network designs.

Based on the analysis of these 5 datasets, the following "no 3's and no 8's" design rule is proposed for mitigating kinetic dispersion caused by unnecessary duplexes. This design rule has two conditions: (1) the network should contain no unnecessary intra-oligomer duplex composed of 3 or more contiguous base-pairs, and (2) the network should contain no unnecessary inter-oligomer duplex composed of 8 or more base-pairs.

The logic underlying the "no 3's and no 8's" design rule is as follows. For intra-oligomer duplexes, evidence was observed that some unnecessary 3-base-pair intra-oligomer duplexes are substantial enough to affect hybridization kinetics under typical experimental conditions. Thus, it is recommended that all unnecessary intra-oligomer duplexes of 3 or more base-pairs be removed. For inter-oligomer duplexes, no clear evidence of unnecessary inter-oligomer duplexes affecting hybridization kinetics was observed, even though all 5 datasets contained such duplexes 7 base-pairs or larger. Since it is well established that large inter-oligomer duplexes can affect hybridization kinetics[9,23,36–40], it is conservatively recommended that unnecessary inter-oligomer duplexes larger than 7 base-pairs be eliminated.

**Scaling of unnecessary duplexes with network size.** For networks of DNA oligomers, it is known that the largest unnecessary duplex generally increases with both the number and length of oligomers present[38,53,54]. Here, the scaling of unnecessary duplexes with these factors was studied using in-silico random sampling. Networks were randomly sampled for the model system design shown in Fig. 2a, which contained: (1) $i$ oligomers, (2) $j$ bases per oligomer, and (3) no intentional base-pairing. Since each of these networks contain no intentional duplexes, $\Delta N = N$ and $\Delta O = O$ for each network.

Values of $\Delta N$ and $\Delta O$ estimated for several $i$ and $j$ combinations are shown in Fig. 2b,c. In these figures, only data for select $j$ values (i.e., 16, 64, 256, and 1024 bases) are shown. Additional $j$ values (i.e., 8, 32, 128, and 512 bases) were used for fitting but are omitted to reduce clutter in the figure. Both $\Delta N$ and $\Delta O$ appear to scale exponentially with both $i$ and $j$. This was modeled using the equations $\Delta N = a \cdot i^b \cdot j^c$ and $\Delta O = a \cdot i^b \cdot j^c$. In these equations, $a$, $b$, and $c$ are real-valued constants. The units of constant $a$ are fitness points and constants $b$ and $c$ are both unitless. $\Delta N$ values were well modeled by values of $a = 0.389$, $b = 3.19$, and $c = 3.68$ (green lines in Fig. 2b) and $\Delta O$ values were well modeled by values of $a = 0.0526$, $b = 1.56$, and $c = 3.63$ (orange lines in Fig. 2c). Both $\Delta N$ and $\Delta O$ values were observed to systematically diverge from this model for small $j$ values, which can be explained by the upper limit oligomer length establishes on unnecessary duplex length.

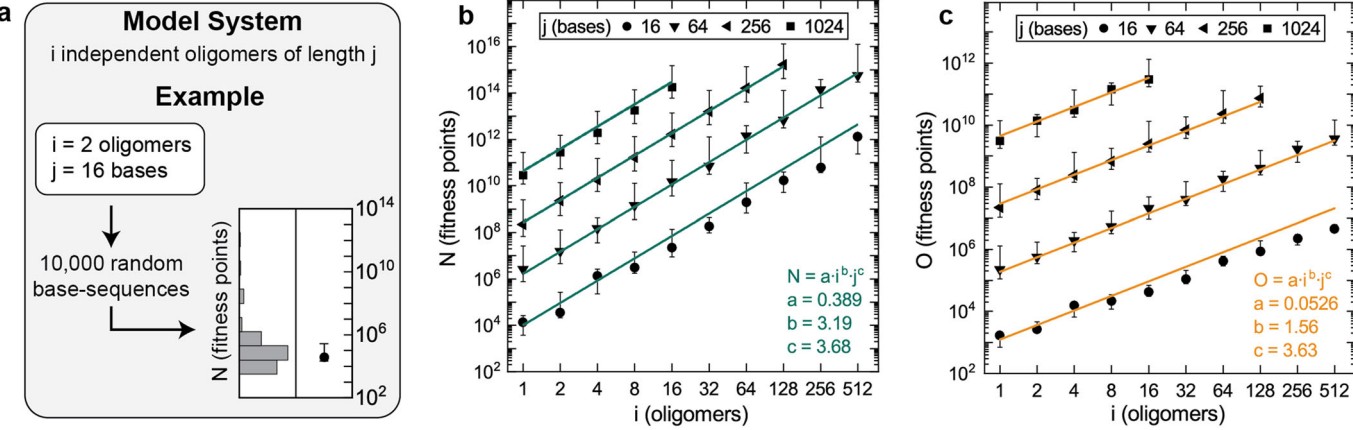

**Fig. 2 In-silico characterization of typical unnecessary duplexes. a** The model system used for this study. **b, c** Typical N (network fitness score) and typical O (oligo fitness score) of randomly sampled networks for select combinations of i and j. Each lower error bar, data point, and upper error bar represent the 25th, 50th, and 75th percentile of 10,000 oligomer-sets. Colored lines and inset equations represent the fits to the data.

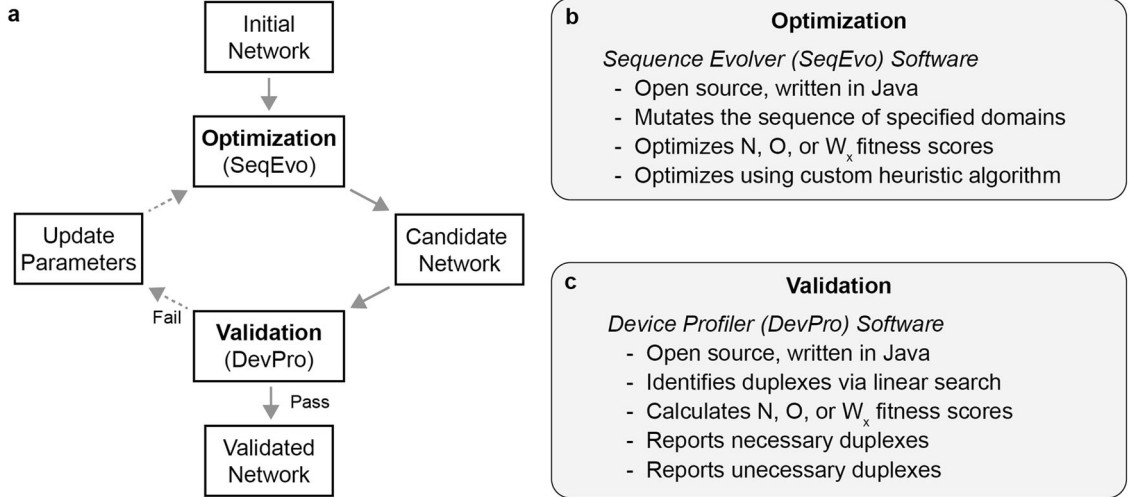

**Fig. 3 Method used to generate optimized networks. a** Illustration of the method. **b** Key details pertaining to the SeqEvo optimization program. **c** Key details pertaining to the DevPro analysis program.

It was inferred from this data that nearly all networks possess unnecessary duplexes substantial enough to cause kinetic dispersion. This is based on the following logic. First, during the analysis of existing experimental data (Fig. 1), 3-base-pair unnecessary intra-oligomer duplexes were observed to affect hybridization kinetics under typical experimental conditions. When the length of the largest intra-oligomer unnecessary duplex is directly calculated for randomly sampled networks, 50% of networks contain 3 base-pair intra-oligomer unnecessary duplexes when: (1) the network contains any oligomer 16 bases or larger, or (2) the network contains any 16 oligomers longer than 8 base-pairs. Networks satisfying these conditions are a small number of the countably infinite number of possible networks, implying that nearly all networks possess unnecessary duplexes substantial enough to affect hybridization kinetics.

This can also be inferred using the following logic. According to the definition of $\Delta O$, a single 3 base-pair unnecessary intra-oligomer duplex contributes $10^3$ fitness points to $\Delta O$. Since $\Delta O$ increases with both $i$ and $j$, networks with $\Delta O \leq 10^3$ are only a small fraction of the countably infinite number of possible networks. Thus, nearly all networks possess unnecessary duplexes substantial enough to affect hybridization kinetics. The model in

Fig. 2c implies that this is true for over 50% of networks when $(0.0526) \cdot i^{1.56} \cdot j^{3.63} \geq 10^3$. This condition is satisfied by networks containing any of the following: (1) an oligomer 16 bases or longer, (2) any 2 oligomers longer than 11 bases, or (3) any 4 oligomers longer than 8 bases. Based on this logic, the number of 8 base-pair oligomers that contain substantial unnecessary duplexes is approximately ¼ that of the prior logic, which can be explained by the fitness points accumulated from 2 base-pair unnecessary intra-oligomer duplexes.

**Generation of optimized networks using the SeqEvo and DevPro software.** Networks forming less substantial unnecessary duplexes were generated using the following process which is depicted graphically in Fig. 3a. At this level of abstraction, the process begins with formalizing a design (i.e., the oligomers and intentional duplexes) for the network. An initial implementation of this design, referred to as the initial network, is created as a starting point for optimization. A first round of optimization is then performed using default parameters, yielding the first candidate network. This candidate network is evaluated in greater detail and either accepted as final or rejected. If rejected, either

the optimization parameters or the network design are updated and the optimization/validation cycle is repeated until a candidate network successfully passes validation.

For the optimization step shown in Fig. 3a, the custom-written Sequence Evolver (abbreviated SeqEvo) computer program was created. This program, instructions for its use, and several usage examples are freely available online[55]. SeqEvo is a command line tool written in the java programming language. As input, the program requires four files: (1) a parameters file, (2) a fixed-domains file, (3) a variable-domains file, and (4) an oligomers file. The parameters file specifies runtime parameters such as which fitness score to optimize. The fixed-domains file specifies a list of named base-sequences, referred to as fixed domains, which will not be modified during optimization. The variable-domains file specifies a list of named base-sequences, referred to as variable domains, which will be modified during optimization. The oligomers file specifies a list of named oligomers, with each oligomer declared by listing one or more domains and/or domain complements. The SeqEvo program compiles the initial network from these files and proceeds with optimization. Optimization is performed using a custom evolution-inspired heuristic algorithm which rearranges the bases within variable domains to minimize a given fitness score. There are 5 key parameters which control the optimization algorithm (labeled CPL, GPC, NDPM, NL, and NMPC) and default values for these parameters (CPL = 100000, GPC = 1, NDPM = 1, NL = 8, and NMPC = 1) were determined by maximizing algorithm efficiency for a model system. At the end of the optimization process, SeqEvo outputs a report file detailing the most-fit network encountered and the runtime parameters used for optimization. Supplementary note 2 reports additional details regarding the operation of SeqEvo. This includes pseudocode explaining the optimization algorithm (supplementary fig. S13) and details of the process used to identify default parameter values (supplementary table S3 and supplementary fig. S14).

For the validation step shown in Fig. 3a, the custom-written Device Profiler (abbreviated DevPro) software was created. This program, instructions for its use, and several usage examples are freely available online[55]. Similar to SeqEvo, DevPro is a command line tool written in the java programming language. The program requires the same four input files (i.e., a parameters file, a fixed-domains file, a variable-domains file, and an oligomers file). DevPro compiles the network and uses an exhaustive linear search to identify all necessary and unnecessary duplexes. The program then outputs a report summarizing these duplexes. If requested in the parameters file, the program can also output values for select fitness scores and a list of the largest unnecessary duplexes. Supplementary note 3 reports additional details regarding the DevPro program. This includes pseudocode for the algorithms used to identify duplexes (supplementary figs. S15 and S16). This also includes example score calculations (supplementary figs. S17 and S18).

As of writing, SeqEvo and DevPro have the following requirements: (1) Both programs require Java SE 8 or newer to be installed and (2) both programs require that the network's design be described using binding domains. Correct function of both programs has been verified on computers running Windows, Linux, and MacOS operating systems. Both programs require more time and memory for larger network designs, however no absolute limits were encountered. The largest network analyzed using the current version of the programs is the "10 x 10 x 10 molecular canvas" reported by Ke et al.[56], which consists of 517 total oligomers containing up to 48 bases per oligomer.

If default parameters fail to generate a network of sufficient quality, it is recommended that future researchers tune the following parameters. (1) The CPL parameter. The CPL parameter controls the number of evolutionary cycles performed during optimization. Increasing CPL increases the duration of the heuristic algorithm, allowing more mutations to be considered and more evolved networks to be generated. (2) The intraSLC and interSLC parameters, which abbreviate the intra-oligomer scoring length criterion and the inter-oligomer scoring length criterion, respectively. Unnecessary duplexes less than intraSLC or interSLC do not contribute points to fitness scores. Thus, increasing these values allows the optimization algorithm greater freedom to reduce larger duplexes by introducing unpenalized smaller duplexes. (3) The maxAA, maxCC, maxGG, and maxTT parameters. These parameters control the maximum number of consecutive bases which are acceptable in generated networks. The default limits of 6, 3, 3, and 6 were set based on the observations of previous researchers[31,57,58]. However, these values substantially limit the number of potential networks, especially for larger network designs. Increasing these parameters allows the optimization algorithm greater freedom to eliminate unnecessary duplexes but may result in adverse experimental outcomes. (4) The scoringWeightX parameter. The scoring-WeightX parameter controls the value of x for the $W_x$ fitness score. Increasing the scoringWeightX parameter increases the contribution of intra-oligomer duplexes to the $W_x$ score, which allows greater freedom to reduce intra-oligomer duplexes by increasing inter-oligomer duplexes.

The following key observations regarding this generation method were made. (1) This generation method treats unnecessary duplexes as fully independent and neglects any cooperation between multiple duplexes. It is possible that the effects observed here do not arise from a single unnecessary duplex, but instead from the cooperative effects of multiple unnecessary duplexes. (2) Both the "no 3's and no 8's" design rule and the fitness scores used here quantify unnecessary duplexes based on their size (i.e., the number of base-pairs they contain). This is derived from the idea that all duplexes of a given length are equally bad, which becomes a poor assumption for duplexes containing approximately 4 or more base pairs since some 4 base-pair duplexes are more thermodynamically stable than some 5 base-pair duplexes. Consequently, when larger unnecessary duplexes are present, these metrics may favor shorter duplexes which are actually more problematic. However, applying some form of thermodynamic criteria to distinguish between unnecessary duplexes appears complicated by the fact that it has been specifically reported that some thermodynamically unfavorable duplexes can affect hybridization kinetics[22]. (3) The fitness scores used here are inefficient in the sense that they penalize all inter-oligomer and all intra-oligomer unnecessary duplexes equally, while not all unnecessary duplexes appear to affect all hybridization reactions. Consequently, these scores likely overlook many experimentally viable networks where the unnecessary duplexes are either not stable enough or located in locations which do not affect the intended hybridization reaction. (4) There is an underlying logical link between intra-oligomer duplexes and inter-oligomer duplexes such that any intra-oligomer duplex implies the existence of two equally sized inter-oligomer duplexes. It is possible that effects attributed to unnecessary intra-oligomer duplexes can also be explained by these inter-oligomer duplexes.

**Kinetic dispersion of newly generated networks**. To further investigate the relationship between unnecessary duplexes and kinetic dispersion, in-vitro kinetic dispersions were measured for several newly generated networks. The model system for this study is shown in Fig. 4a and consists of three oligomers (labeled $S_1$, $S_2$, and $S_3$) which are intended to undergo two separate

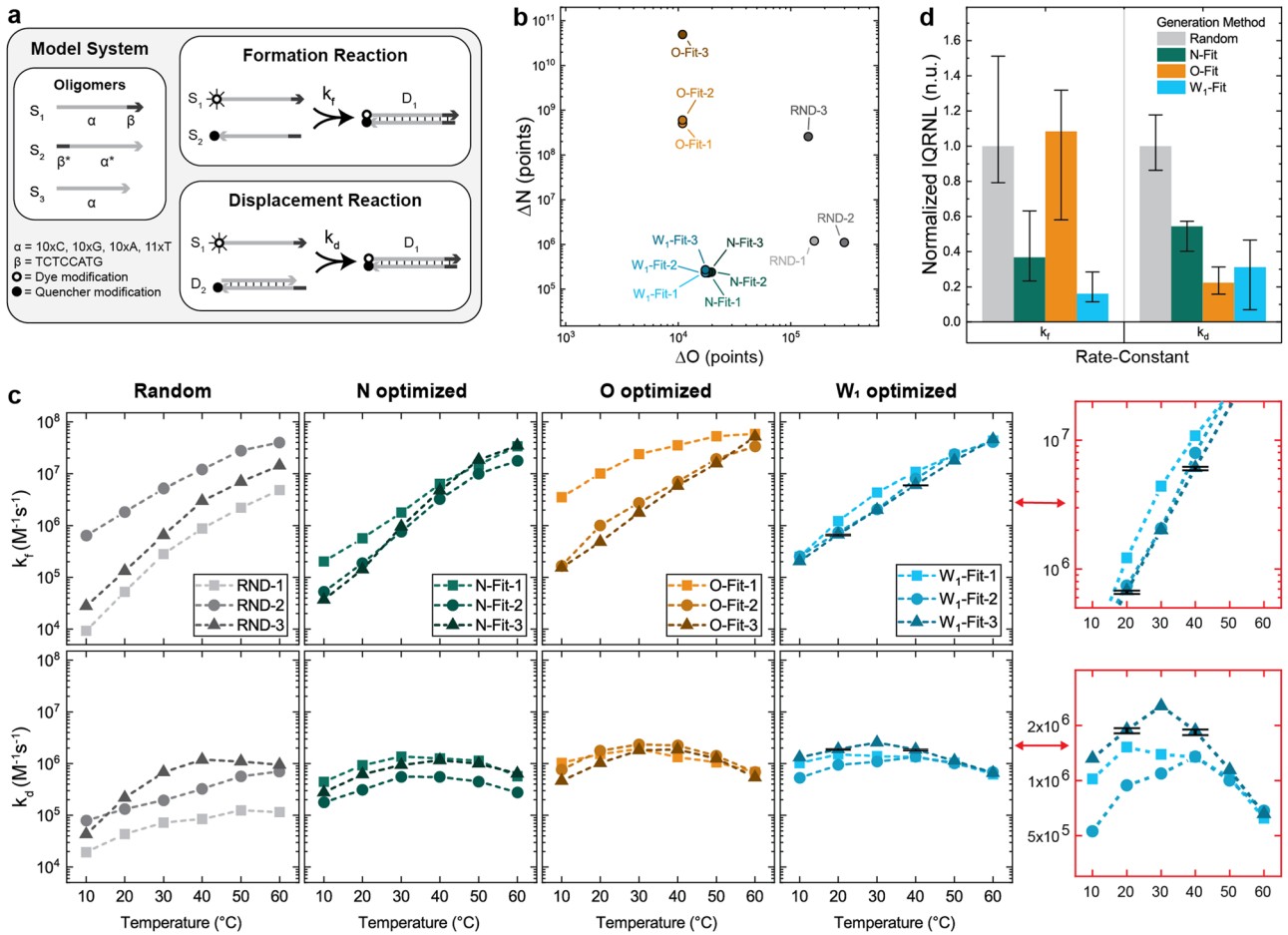

**Fig. 4 In-vitro characterization of newly generated oligomers. a** The design of the model system for this study. DNA oligomers are represented by lines with arrows indicating the 3′ end of the molecule. Shading indicates the two binding domains (α, β) and their complements (α*, β*). This design is intended to undergo two hybridization reactions: formation (modeled using rate constant $k_f$) and displacement (modeled using rate constant $k_d$). **b** Fitness scores of the twelve new networks generated for the model system. **c** Rate-constant values measured in-vitro for the twelve networks. Dashed lines are shown connecting discrete data points. Error bars indicate the mean and standard deviation of triplicate measurements for select data points. These error bars are emphasized in the two red regions expanded to the right. **d** Kinetic dispersions estimated for each network generation method. The lower error bar, the column height, and the upper error bar represent the 25th, 50th, and 75th percentiles of IQRNL estimates.

hybridization reactions (labeled duplex-formation and strand displacement).

New networks were generated using one of four network generation methods: (1) random sequence assignment (the RND group), optimization of the N fitness score (the N-fit group), optimization of the O fitness score (the O-fit group), or optimization of the $W_1$ fitness score ($W_1$-fit group). Three networks were generated using each generation method. This number of samples was chosen based on the magnitude of previously observed kinetic dispersion reductions. For $W_x$ optimization, the value of $x = 1$ was chosen because it was the smallest x value which satisfied the "no 3's and no 8's" design rule. Supplementary note 4 reports additional details regarding the generation and characterization of these networks. This includes the base-sequence for each oligomer, which are reported in supplementary table S4.

The unnecessary duplexes for each network were characterized using the DevPro program. Values of ΔN and ΔO for each network are shown in Fig. 4b. Histograms summarizing the unnecessary duplexes for each network are shown in supplementary fig. S19. Networks in the RND group contain unnecessary intra-oligomer duplexes as large as 5 base-pairs and unnecessary inter-oligomer duplexes as large as 8 base-pairs. No N-fit, O-fit,

or $W_1$-fit network contained unnecessary intra-oligomer duplexes larger than 2 base-pairs and no N-fit or $W_1$-fit network contained unnecessary inter-oligomer duplexes larger than 4 base-pairs. The ΔO > 10,000 exhibited by the N-fit, O-fit, and W1-fit networks result from the existence of numerous small intra-oligomer duplexes. For example, the N-fit-1 network's ΔO value of 18,400 is the sum of points from 790 1-base-pair duplexes and 105 2-base-pair duplexes. Relative to the other networks, O-fit networks contained relatively large unnecessary inter-oligomer duplexes, including inter-oligomer duplexes of up to 10 base-pairs. This indicates a logical connection between inter-oligomer and intra-oligomer duplexes such that networks with less substantial intra-oligomer duplexes tend to possess more substantial inter-oligomer duplexes.

The chemical kinetics of each network were characterized using a fluorescence signal and modeled using second-order rate equations. The duplex-formation reaction was modeled using the following rate-equation governed by rate constant $k_f$.

$$S_1 + S_2 \xrightarrow{k_f} D_1 \tag{1}$$

$$\frac{d[D_1]}{dt} = k_f [S_1][S_2] \tag{2}$$

The strand-displacement reaction was modeled using the following rate-equation governed by rate-constant $k_d$:

$$S_1 + D_2 \xrightarrow{k_d} D_1 + S_3 \tag{3}$$

$$\frac{d[D_1]}{dt} = k_d [S_1][D_2] \tag{4}$$

Based on these equations, the rate-constants $k_f$ and $k_d$ both have units of $M^{-1}s^{-1}$.

Rate-constants were characterized at several experimental temperatures, and the measured rate-constant values are shown in Fig. 4c. The largest rate-constant value observed was $5.9 \times 10^7$ $M^{-1}s^{-1}$ ($k_f$, O-fit-1, 60 °C), and the smallest was $9.2 \times 10^3$ $M^{-1}s^{-1}$ ($k_f$, RND-1, 10 °C). The range of these rate-constant values is consistent with values reported elsewhere in the literature[12,22,23,43,47,59–63]. The mean and standard deviation of triplicate measurements for select data points are shown as error bars on Fig. 4c. The maximum difference within these triplicate measurements was 4% or less. Reports detailing the measured rate-constant values and associated fluorescence data are indexed in supplementary table S5 and shown in supplementary figs. S20 to S95.

Network generation via $W_1$ optimization yielded the lowest kinetic dispersions. Across the six experimental temperatures, these kinetic dispersions were reduced by a median of 86% (formation reaction) and 75% (displacement reaction) relative to the randomly generated networks. These reductions are consistent with those observed while analyzing existing experimental data (Fig. 1). Three key pieces of information were inferred from the low kinetic dispersions of the $W_1$-fit networks: (1) Substantial kinetic dispersion remained despite the factors held constant in this study (*e.g.*, temperature, viscosity, ionic strength, oligomer length, duplex stability, and toehold sequence), (2) unnecessary duplexes contributing to $W_1$ explain most of this kinetic dispersion, and (3) reducing unnecessary duplexes using *in-silico* optimization of $W_1$ substantially mitigated kinetic dispersion for this model system.

Optimizing N yielded networks with unnecessary duplexes very similar to the $W_1$-fit group. However, the $k_f$ for two of the N optimized networks (N-Fit-2 and N-fit-3) differed substantially from the $W_1$-fit group. This effect was especially pronounced at low temperatures, where rate-constants differed up to a factor of 7. This kinetic dispersion can be explained by unnecessary intra-oligomer duplexes containing 2 base-pairs. This is consistent with the prior evidence that 2-base-pair unnecessary intra-oligomer duplexes are substantial enough to affect hybridization kinetics under some conditions. It was inferred from this data that the kinetic dispersion of 2-base-pair unnecessary intra-oligomer duplexes were negligible above approximately 30 °C.

Optimizing O yielded two networks with kinetics similar to the $W_1$-fit group, and one network with a substantially faster duplex-formation reaction. This accelerated reaction may be explained by the numerous large unnecessary inter-oligomer duplexes introduced during O optimization (i.e., duplexes as large as 8 base-pairs). Notably, the two O-fit networks with kinetics similar to the $W_1$-fit group also contain large unnecessary inter-oligomer duplexes (i.e., as large as 10 base-pairs) which indicates that not all unnecessary inter-oligomer duplexes affect all hybridization reactions.

For all optimized networks, kinetic dispersion decreased at higher temperatures. This effect was especially pronounced for the $W_1$-fit oligomers undergoing the displacement reaction at 50 °C and 60 °C. At these temperatures, kinetic dispersions were reduced by 94% and 96% relative to randomly generated oligomers. The remaining kinetic dispersion for the $W_1$-fit oligomers at these temperatures appears to be beyond the resolution of the current study, which was estimated at 4%. The trend of decreasing kinetic dispersion with increasing temperature may be explained by the destabilization of smaller intra-oligomer unnecessary duplexes. From this, it was inferred that operating networks of DNA oligomers at relatively high temperatures may help mitigate kinetic dispersion, especially for networks containing relatively small unnecessary duplexes.

**Temperature dependence of hybridization kinetics**. Rate-constants for the newly generated networks were measured at temperatures of 10, 20, 30, 40, 50 and 60 °C. In this temperature range, rate constants for the duplex-formation reaction were observed to be strictly increasing, suggesting an Arrhenius temperature dependence. Similar reactions where oligomers form a single duplex have been characterized in previous studies and both Arrhenius[41,43,47,64] and non-Arrhenius[22,47,65] temperature dependences have been observed. Alternatively, rate constants for the strand-displacement reaction typically exhibited a broad peak with a maximum rate-constant between 20 and 40 °C. The decreased strand-displacement rate-constants at higher temperatures can be explained by a destabilizing intermediate, which was presumed to be a three-oligomer complex prior to strand-displacement.

The temperature dependence of duplex-formation rate-constants were modeled using the following equation:

$$k_f = A \cdot \exp\left(\frac{-E_a}{k_B T}\right) \tag{5}$$

Referred to as the Arrhenius equation, this equation includes the following values: the pre-exponential factor (A), the activation energy ($E_a$), the absolute temperature (T), and the Boltzmann constant ($k_B$). Based on Eq. 5, a plot of the natural logarithm of $k_f$ as a function of inverse temperature is expected to be linear, and such plots are shown in Fig. 5a. The activation energy and pre-exponential factor extracted from these fits are shown in Fig. 5b.

The duplex-formation rate-constants of all twelve networks were well described by the Arrhenius equation, which models the rate-limiting step of a reaction as a constant energy barrier overcome using thermal energy. Dispersion in the magnitude of this barrier (*i.e.*, the activation energy $E_a$) followed similar trends to kinetic dispersion and was most uniform for the $W_1$ optimized networks. A plot of the natural logarithm of the pre-exponential factor as a function of activation energy was observed to be highly linear and is shown in Fig. 5c. This implies the following equation relating these parameters:

$$\ln(A) = C_1 + C_2 E_a \tag{6}$$

where $C_1$ and $C_2$ are real valued constants. A linear fit to the data in Fig. 5c is shown as a solid line. This fit yielded $C_1$ and $C_2$ values of 19.1 log $M^{-1}s^{-1}$ and $20.6 \times 10^{19}$ $J^{-1}$ log $M^{-1}s^{-1}$, respectively. Equation 6 was simplified by declaring the following constants:

$$C_3 \equiv \exp(C_1) \tag{7}$$

$$T_c \equiv \frac{1}{k_B C_2} \tag{8}$$

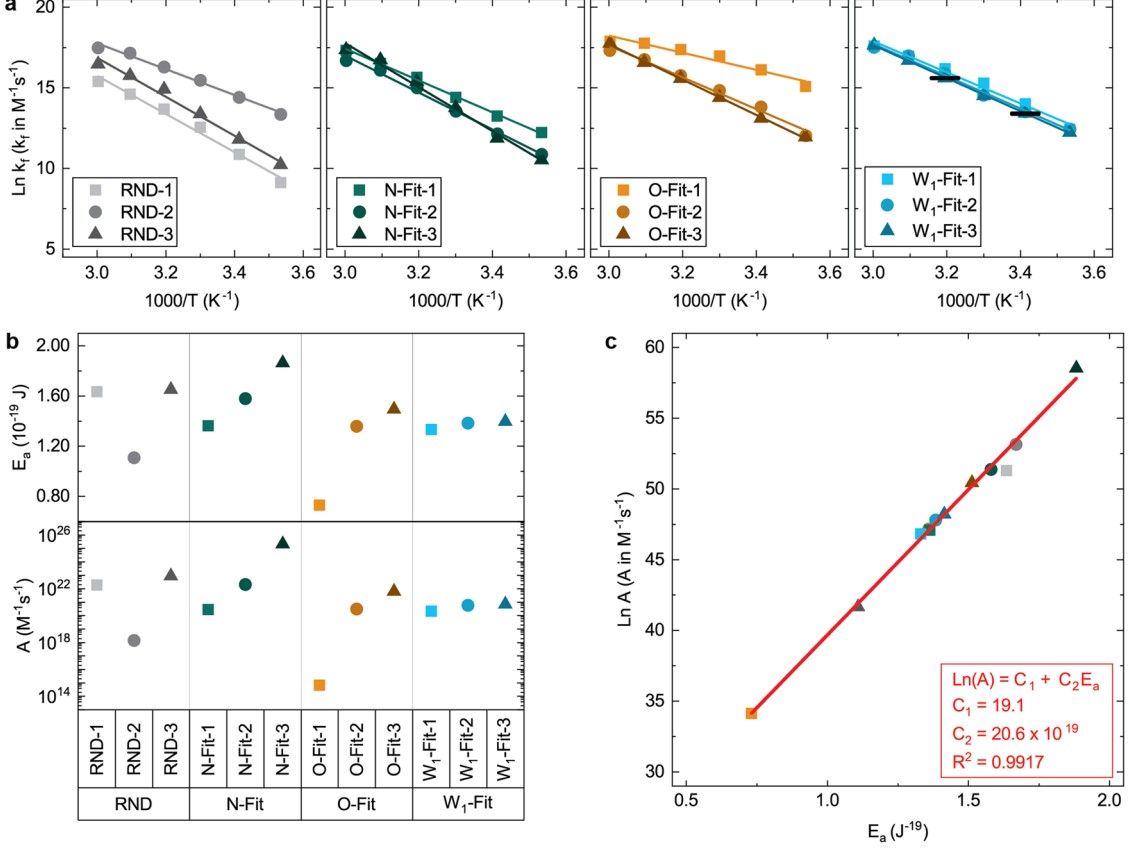

**Fig. 5 Arrhenius temperature dependence of $k_f$. a** Arrhenius plots of the same $k_f$ values reported in Fig. 4. Linear fits are shown as solid lines. Error bars indicate the mean and standard deviation of the same triplicate measurements in Fig. 4. **b** The activation energy ($E_a$) and pre-exponential factor ($A$) associated with the linear fits. **c** The correlation between activation energy and pre-exponential factor modeled using a linear fit (red line, inset equation).

which yielded the following empirical model for $k_f$:

$$k_f = C_3 \cdot \exp\left(\frac{E_a}{k_B} \cdot \left(\frac{1}{T_c} - \frac{1}{T}\right)\right) \quad (9)$$

One feature of this equation is the theoretical critical temperature ($T_c$) at which formation rates equal the maximum rate ($C_3$) regardless of the value of the activation energy ($E_a$). The $C_1$ and $C_2$ values from the fit to Eq. 6 imply $C_3$ and $T_c$ values of $1.97 \times 10^8$ M⁻¹s⁻¹ and 352 K. These values are likely specific to the duplex-formation reaction in Fig. 4a and factors such as viscosity, ionic strength, oligomer length, and duplex stability.

Correlations such as Eq. 6 have been observed by previous researchers and were interpreted as evidence of enthalpy/entropy compensation in an underlying thermodynamic model[66]. However, this explanation is contradicted by evidence that duplex formation reactions are a dynamic equilibrium governed by a rate-limiting transition through an intermediate[42,61,67,68]. As such, it was concluded that Eq. 9 arises from a similarity in the rate-limiting reaction mechanism of the reactions.

These results suggest the following model of how unnecessary duplexes affect hybridization reactions. Hybridization reactions exhibit highly similar chemical kinetics manifesting as low kinetic dispersions when all known factors, including unnecessary duplexes, are sufficiently controlled. For hybridization reactions where a single duplex is formed, these chemical kinetics exhibit an Arrhenius temperature dependence. Relative to these chemical kinetics, unnecessary duplexes can either accelerate or decelerate a given hybridization reaction, which manifests as increased kinetic dispersion. Different unnecessary duplexes affect the

reaction mechanism in different ways, leading to a divergence in reaction mechanisms and many possible behaviors. This model appears to adequately describe both relatively simple reactions such as the duplex-formation reaction in Fig. 4a, and more complicated reactions such as those from Fig. 1. For at least some hybridization reactions, these divergent reaction mechanisms continue to follow mathematical relationships such as Eq. 9, regardless of the effects of unnecessary duplexes.

**Generation of largest possible networks**. Numerous methods for generating DNA oligomers have been reported and there exist at least fifteen computer programs specifically intended to generate networks of interacting DNA oligomers[7,38,53,54,57,69–78]. The size limitations of SeqEvo and several freely available programs were assessed *in-silico* by generating the largest possible networks obeying the "no 3 s and no 8 s" rule. New networks were generated for two separate model systems. The first model system consisted of a single intentional duplex containing the maximum possible number of base-pairs. The second model system consisted of the maximum possible number of independent 8-base-pair duplexes. The size of the largest networks generated for these model systems is reported in Fig. 6.

The largest networks yielded by default parameters of the SeqEvo program were a single 560-base-pair duplex and a network of 464 8-base-pair duplexes. For comparison, assigning bases randomly yielded only a 12-base-pair duplex and a network of 4 8-base-pair duplexes, indicating a significant improvement in network size. Relative to the other generation methods, the default parameters of SeqEvo yielded both the second largest

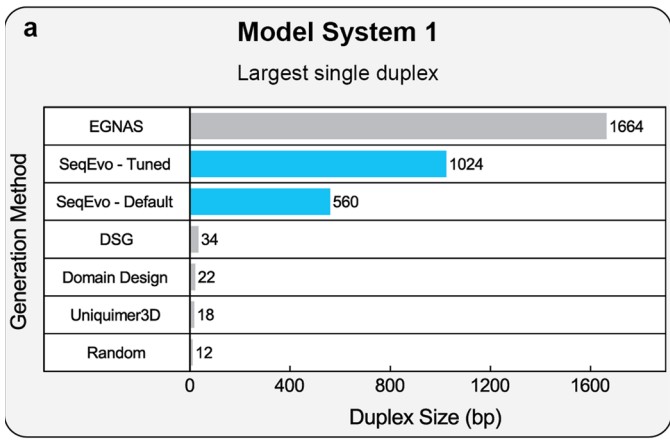

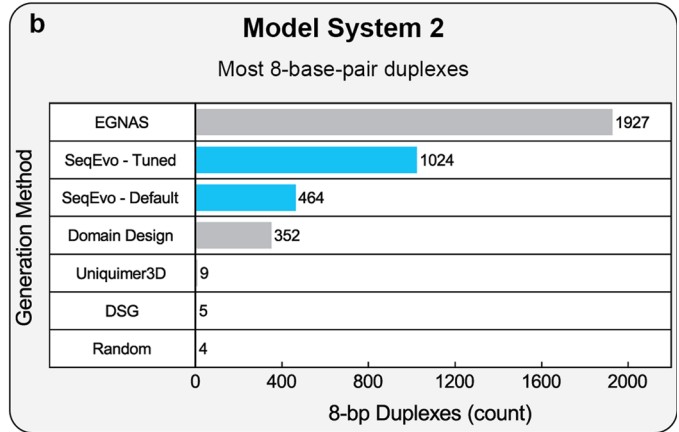

**Fig. 6 In-silico evaluation of generation method limits. a** Size of the largest single duplex generated obeying the "no 3's and no 8's" design rule. **b** Size of the largest network of 8-base-pair duplexes generated obeying the "no 3's and no 8's" design rule.

**Table 1 Summary of unnecessary duplex reductions for existing network designs.**

| Network | $\Delta N$ | $\Delta O$ | Largest intra [a] | Largest inter [b] |
|---|---|---|---|---|
| Autocatalytic Network (Zhang et al.[59]) | | | | |
| As published | $1.32 \times 10^6$ | $7.50 \times 10^4$ | 4 | 4 |
| $W_{100}$ Optimized | $4.10 \times 10^5$ | $1.51 \times 10^4$ | 2 | 3 |
| Autocatalytic Network (Kotani et al.[79]) | | | | |
| As published | $1.11 \times 10^{45}$ | $6.35 \times 10^5$ | 5 | 45 |
| $W_{1,000,000}$ Optimized | $3.97 \times 10^7$ | $9.78 \times 10^4$ | 2 | 6 |
| Four-Input OR Network (Qian et al.[58]) | | | | |
| As published | $2.71 \times 10^{22}$ | $8.08 \times 10^4$ | 4 | 22 |
| $W_{10,000}$ Optimized | $3.13 \times 10^{10}$ | $5.18 \times 10^4$ | 2 | 9 |
| 10x10x10 Canvas (Ke et al.[56]) | | | | |
| As published | $2.22 \times 10^{25}$ | $2.15 \times 10^8$ | 8 | 25 |
| $W_{100,000,000}$ Optimized | $1.96 \times 10^{11}$ | $1.94 \times 10^6$ | 3 | 10 |

[a]The largest-intra column reports the number of base-pairs in the largest unnecessary intra-oligomer duplex.
[b]The largest-inter column reports the number of base-pairs in the largest unnecessary inter-oligomer duplex.

single duplex and the second largest network of 8-base-pair duplexes.

The EGNAS program yielded both the largest single duplex and the largest network of 8-base-pair duplexes. However, the EGNAS program is not currently capable of generating oligomers for other network designs. The large networks generated using this program were attributed to three primary factors: (1) EGNAS appears specifically optimized for this type of network design (i.e., networks with no intentional connections between duplexes), (2) EGNAS quantifies unnecessary duplexes based on their length in base-pairs, and (3) the authors of EGNAS included clearly labeled parameters controlling the size of the largest intra-oligomer unnecessary duplex and the size of the largest inter-oligomer duplex. These factors collectively make EGNAS well suited for generating networks of independent duplexes obeying the "no 3's and no 8's" design rule.

The default parameters of the NUPACK software yielded no networks satisfying the "no 3's and no 8's" design rule. This was attributed to the following factors: (1) NUPACK differentiates between networks using thermodynamic simulation and not based on the number of base-pairs in unnecessary duplexes, and (2) NUPACK optimizes for network thermodynamics, and relatively small unnecessary intra-oligomer duplexes rarely affect the thermodynamic stability of intentional duplexes. It has been specifically reported that some thermodynamically unfavorable duplexes can affect hybridization kinetics[22].

However, it is possible that the unnecessary duplexes present in NUPACK-generated networks have no impact on hybridization kinetics. A future study directly comparing the kinetic dispersion of networks generated using thermodynamic and non-thermodynamic optimization criteria may provide valuable insight into the relationship between unnecessary duplexes and reaction kinetics.

Each generation method studied here has parameters which can be changed by advanced users, and a sufficiently advanced user could almost certainly tune these parameters to generate larger networks satisfying the "no 3's and no 8's" design rule. For example, a network of 1024 8-base-pair duplexes was generated using the SeqEvo software by tuning the following parameters: (1) CPL increased from 100,000 to 200,000. (2) interSLC increased from 1 to 8. (3) intraSLC increased from 1 to 3. A single 1024 base-pair duplex satisfying the "no 3's and no 8's" design rule was generated from the SeqEvo software by tuning the following parameters: (1) CPL increased from 100,000 to 1,000,000. (2) intraSLC increased from 1 to 3. (3) interSLC increased from 1 to 8. Developing optimal methods for mitigating kinetic dispersion using SeqEvo or other programs is a promising opportunity for future research and may substantially increase the size of networks for which kinetic dispersion can be mitigated.

**New oligomers for existing designs**. A key advantage of SeqEvo relative to other freely available design tools is this program's

ability to generate networks for a variety of different network designs. However, larger and/or more complicated network designs may introduce issues that limit the effectiveness of this program. To better understand the variety of networks which can be generated using SeqEvo, new networks were generated for several existing network designs.

Attempts were made to generate networks satisfying the "no 3's and no 8's" design rule for 4 existing network designs. These designs included: (1) the "entropy driven autocatalytic network" reported by Zhang et al.[59], (2) the "autocatalytic four-arm junction" reported by Kotani et al.[79], (3) the "four-input OR seesaw-gate network" reported by Qian et al.[58], and (4) the "10 x 10 x 10 molecular canvas" reported by Ke et al.[56]. The DevPro software was used to characterize the unnecessary duplexes for both published and newly generated networks. These unnecessary duplexes are summarized in Table 1. Supplementary note 5 reports additional information regarding these networks. The base-sequence of the new networks and the number of unnecessary duplexes of each length are detailed in supplementary tables S6 to S13.

Unnecessary duplexes were successfully reduced for each of the designs, however the "no 3's and no 8's" design rule could not be satisfied for two of the four designs. The failure to generate a satisfactory network for the "10 x 10 x 10 molecular canvas" was attributed to the size of this design, which contained 517 total oligomers and oligomers up to 48 bases in length.

The amount of computation required to generate the new networks depended strongly on network design. The least amount of computation was required by the "entropy-driven autocatalytic network" design. The new network for this design was generated after scoring 800,000 total networks in 2 min using a laptop computer (Intel Xeon E3-1505M v5 processor). The most computation was required for the "10 x 10 x 10 molecular canvas" design. The new network for this design was generated after scoring 40,000,000 networks using a single node of a computing cluster for 28 h (dual intel Xeon E5-2680 v4 processors). The following SeqEvo parameters were tuned to generate this new network: (1) CPL increased from 100,000 to 1,000,000. (2) intraSLC increased from 1 to 3. (3) interSLC increased from 1 to 5. (4) scoringWeightX increased from 10,000 to 100,000,000.

## Discussion

For networks of interacting DNA oligomers, it is known that stretches of Watson-Crick base-pairing which are not part of an intended hybridization reaction, here referred to as unnecessary duplexes, can affect the chemical kinetics of hybridization reactions[9,22,23,36–40]. However, these chemical kinetics also depend on other known factors such as temperature[12,22,41–44], ionic strength[45,46], viscosity[45,47], oligomer length[42], and duplex stability[12,22,48,49]. Here, a difference in chemical kinetics is referred to as kinetic dispersion, and the relationship between unnecessary duplexes and kinetic dispersion was studied.

Several key observations were made regarding unnecessary duplexes and kinetic dispersion. For one, evidence was observed that some unnecessary intra-oligomer duplexes containing as few as 2 base-pairs are substantial enough to cause kinetic dispersion under certain conditions. Such evidence was observed both while analyzing previously reported in-vitro kinetic dispersions (Fig. 1) and while analyzing the in-vitro kinetic dispersions of newly generated networks (Fig. 4). Evidence was also observed that not all unnecessary duplexes affect all hybridization reactions. For the conditions studied here, the effects of 2-base-pair unnecessary intra-oligomer duplexes appear mitigated at temperatures of 30 °C or higher. However, some 3-base-pair unnecessary intra-oligomer duplexes appear substantial enough to cause kinetic dispersion for a

larger range of experimental conditions. Finally, when networks were randomly sampled in-silico, it was observed that unnecessary duplexes typically scale exponentially with network size (Fig. 2) and that that all but the smallest networks contain unnecessary duplexes substantial enough to cause kinetic dispersion.

Results reported here demonstrate that sufficiently controlling factors known to affect hybridization kinetics can greatly limit kinetic dispersion, and that these reductions are sometimes substantial enough that little to no kinetic dispersion can be observed. For example, networks generated using $W_1$ optimization (W1-Fit group, orange data in Fig. 4) exhibited exceptionally low kinetic dispersions for the duplex-formation reaction at 50 °C. Based on the kinetic model and experimental conditions used here, the fastest and slowest of these reactions should require 4.2 s and 5.6 s to reach half-completion, respectively. Supplementary note 6 details the calculation of these half-completion times and the calculated values are reported in supplementary table S14. When this kinetic dispersion is projected upon an event such as the release of medication from a nanoscale box[14], these 4.2 and 5.6 s half-completion times imply that newly generated boxes may require anywhere between 4.2 to 5.6 s to release one-half of the medication.

Results reported here also indicate that failing to mitigate unnecessary duplexes can lead to substantial kinetic dispersion. For example, new networks generated without in-silico optimization (RND group, grey data in Fig. 4) exhibited notably high kinetic dispersions for the duplex-formation reaction at 10 °C. Based on the kinetic model and experimental conditions used here, the fastest and slowest of these reactions should require 160 and 11,000 s to reach half-completion, respectively. By the same logic above, this implies newly generated boxes may require anywhere between 160 to 11,000 s to release one-half of the medication. While either, or neither, of these time scales may be favorable depending on specifics of the application, not knowing if a newly generated box will release medication in 160 or 11,000 s is almost certainly unfavorable. Such large differences in performance may contribute to problems such as increased development costs or unreliable treatment outcomes. If the narrow range of performances expected for in-silico optimized networks could be replicated in practice, this may reduce the costs associated with treatment development or improve the reliability of treatment outcomes. Other applications for networks of DNA oligomers may also be sensitive to the magnitude of kinetic dispersion observed here[10–13,15–21], and it is plausible that these other applications could similarly benefit from the kinetic dispersion reductions observed here.

The new network generation method reported here utilizes in-silico optimization to mitigate kinetic dispersion caused by unnecessary duplexes. Exceptionally low kinetic dispersions were demonstrated using this method. This included the 3 $W_1$-fit networks (blue in Fig. 4) whose rate-constants at 50 °C differ by no more than a factor of 1.33. Numerous other methods for generating networks of DNA oligomers have been reported[7,38,53,54,57,69–78]. However, to the knowledge of the authors, this is the lowest such kinetic dispersion reported in the literature. Using other reported methods, the lowest kinetic dispersion one could likely achieve would be by generating new networks using a modified version of the rate-constant predicting six-feature weighted neighbor voting algorithm of Zhang et al.[12]. This algorithm was demonstrated to accurately predict most rate-constants within a factor of 2, and it is possible similar kinetic dispersions could be achieved for network generation.

The "no 3's and no 8's" rule is proposed here as a general guideline for researchers looking to mitigate unnecessary duplexes. The following key observations were made regarding this design rule. (1) It is more difficult to satisfy this design rule

for larger networks. While a formal limit for network size was not observed, the largest networks reported here which satisfy this design rule include both a single duplex containing 1664 base-pairs and a network of 1927 8-base-pair duplexes. (2) Networks which satisfy this design rule exhibit reduced kinetic dispersions under a range of typical experimental conditions. However, kinetic dispersions are lowest at experimental temperatures of 30 °C or higher.

To help future researchers mitigate unnecessary duplexes, the SeqEvo and DevPro programs underpinning this network generation method have been made freely available. The following key observations were made regarding these programs. (1) The programs can generate and/or analyze many different network designs. This was specifically demonstrated by generating new oligomers for previously reported network designs (Table 1), for which SeqEvo was able to successfully eliminate 3-base-pair unnecessary intra-oligomer duplexes in 3 of 4 network designs. It was speculated that SeqEvo failed to generate a suitable network for one of these networks due to the size of the design, which contained 517 oligomers and up to 48 bases per oligomer. (2) When the largest possible networks satisfying the "no 3's and no 8's design" were generated using SeqEvo, the program yielded both a single duplex containing 1024 base-pairs and a network of 1024 8-base-pair duplexes. These network sizes were larger than several other freely available design tools and were substantially improved by tuning SeqEvo parameters. The only software which successfully generated networks larger than SeqEvo is not currently capable of generating networks for more complicated designs.

Results here indicate the following promising opportunities for future research. (1) There are likely existing applications for which the "no 3's and no 8's" design rule mitigates kinetic dispersion sufficiently to improve performance and/or reliability. Characterizing the effects of the "no 3's and no 8's" design rule on such applications may both improve the outcomes of these applications and help develop future design rules. (2) There appear to be some designs for which networks satisfying the "no 3's and no 8's" design rule exists but cannot be generated using current software. Improved network generation software may increase the designs for which the "no 3's and no 8's" design rule can be satisfied, which may help expand the applications for which kinetic dispersion can be mitigated. (3) While the "no 3's and no 8's" design rule and the fitness scores utilized here appear highly effective, they were observed to be limited by the assumption that unnecessary duplexes of a given length are equally bad. Development of new fitness scores, design rules and network generation methods could greatly increase the applications for which kinetic dispersion can be mitigated.

## Methods

**IQRNL.** Kinetic dispersion was quantified as the inter-quartile range of the natural logarithm of rate-constant values (abbreviated IQRNL). The following observations regarding this metric were made. Small IQRNL values correspond with smaller, and more favorable, kinetic dispersion. IQRNL values can only be meaningfully compared if they are derived from the same rate-equations and represent the same rate-constant of these equations. IQRNL values are expected to be robust to both occasional outliers and the several orders of magnitude expected for these rate constants. IQRNL values are typically insensitive to the outer 50% of data, making them a conservative metric for quantifying kinetic dispersion. The IQRNL of a sample is expected to systematically underestimate the value of a population, marking it as a biased estimator and making the most meaningful comparisons of IQRNL values those calculated from the same number of samples.

**Datasets.** Datasets which reported 30 or more measurements of a specific rate-constant under consistent experimental conditions were identified for analysis. The five datasets identified include: (1) 47 measurements by Hata et al.[22] of a duplex-formation reaction at room temperature (approximated as 22 °C), (2) 51 measurements by Olson et al.[23] of a catalytic reaction at 25 °C, (3) 51 measurements by Olson et al.[23] of a leak reaction at 25 °C, (4) 98 measurements by Zhang et al.[12] of a duplex-formation reaction at 37 °C, and (5) 95 measurements by Zhang et al.[12] of a duplex-formation reaction at 55°C. The samples within each dataset are expected to have approximately constant temperature, ionic strength, viscosity, oligomer length, and duplex-stability. Either no other oligomers, or only poly-T oligomers were consistently present in each dataset. It is expected that unnecessary duplexes are the only known cause of kinetic dispersion not controlled within each dataset. The oligomers which yielded this data were neither randomly generated nor rationally designed. However, they were approximated as independent and identically distributed[80,81] for the statistical analysis.

**N, O, and W$_x$.** Inter-oligomer duplexes were summarized using the network fitness score (abbreviated N). N was calculated by accumulating $10^L$ fitness points for each duplex connecting two oligomers:

$$N \equiv \sum_{inter-oligo}^{i} 10^{L_i} \tag{10}$$

where L is the number of base-pairs in the duplex. Calculation of N included fitness points for both duplexes which are part of larger duplexes and for each oligomer interacting with an identical oligomer. The value of N is zero when the oligomers can form no inter-oligomer duplexes, and larger values of N indicate more substantial (i.e., larger or more numerous) inter-oligomer duplexes. Unnecessary inter-oligomer duplexes were quantified using the network fitness score above baseline (abbreviated ΔN), which was calculated as network fitness score minus the network fitness score of all necessary duplexes.

Intra-oligomer duplexes were summarized using the oligomer fitness score (abbreviated O). O was calculated by accumulating $10^L$ fitness points for each duplex connecting an oligomer with itself:

$$O \equiv \sum_{intra-oligo}^{i} 10^{L_i} \tag{11}$$

where L is the number of base-pairs in the duplex. Calculation of O included fitness points for duplexes which are part of larger duplexes. The value of O is zero when the oligomers can form no intra-oligomer duplexes, and larger values of O indicate more substantial intra-oligomer duplexes. Unnecessary intra-oligomer duplexes were quantified using the oligomer fitness score above baseline (abbreviated ΔO), which was calculated as oligomer fitness score minus the oligomer fitness score of all necessary duplexes.

Inter-oligomer and intra-oligomer duplexes were collectively summarized using a class of weighted fitness scores (generally abbreviated W$_x$). These fitness scores were calculated as weighted linear combinations of N and O:

$$W_x \equiv N + x \cdot O \tag{12}$$

where x is a positive real number determining the relative contribution of O. Specific W$_x$ are denoted with the value of x in the subscript (e.g., W$_1$ denotes W$_x$ with x = 1). The use of larger x values allows one to place an increasing emphasis on intra-oligomer duplexes. The value of all W$_x$ are zero when no duplexes

are present, and larger values indicate more substantial duplexes. Unnecessary duplexes were quantified using the weighted fitness scores above baseline (abbreviated $\Delta W_x$), which were calculated as the weighted fitness score minus the weighted fitness score of any necessary duplexes.

The following three observations were made regarding these fitness scores. First, N and O are not orthogonal since any intra-oligomer duplex physically implies the existence of two inter-oligomer duplexes. Alternatively stated, if an oligomer can form base-pairs with itself, it can also form these base-pairs with an identical oligomer. Consequently, any structure which contributes to O also contributes to N, and N is always greater than or equal to $2 \cdot O$. Second, while oligomers with $\Delta N$ and $\Delta O$ values approaching zero converge to the design, oligomers with larger values may diverge from the design by a variety of mechanisms resulting in a variety of behaviors. This makes $\Delta N$ and $\Delta O$ appropriate for quantifying unnecessary duplexes, but less appropriate for predicting the behavior of oligomers. Third, DNA oligomers are known to participate in other interactions not captured by N and O, so some oligomers with abnormal behavior may be expected. Such other interactions include non-canonical base-pairing, triplexes, quadruplexes, the binding of oligomers to container walls, or aptamer-like binding to other targets[82,83].

**Estimating kinetic dispersion in existing experimental data.** Fitness score values for each sample were calculated using the custom-written Device Profiler (abbreviated DevPro) computer program. This program utilizes an exhaustive linear search to identify both necessary and unnecessary duplexes. The source code of the DevPro computer program has been made freely available online[55]. Supplementary note 3 provides more information regarding the DevPro program.

The kinetic dispersions of randomly selected samples were estimated using the following process. First, the number of samples to analyze, $n$, was chosen. The rate-constants of $n$ samples were selected randomly. These rate constants were resampled with replacement 1000 times and IQRNL was calculated for each of the resamplings. The 25th, 50th, and 75th percentiles of these IQRNL values were recorded. This process was repeated 1,000 times, and the median of the 25th, 50th, and 75th percentiles are reported in Fig. 1 as the kinetic dispersion estimates.

The kinetic dispersions of the most-fit samples were estimated using the following process. First, the number of samples to analyze, $n$, was chosen. The rate-constants of the $n$ most-fit samples were selected. These rate constants were resampled with replacement 1000 times and IQRNL was calculated for each of these resamplings. The 25th, 50th, and 75th percentiles of these IQRNL values were recorded. Normalized percentiles were calculated by dividing these values by the 50th percentile of the randomly selected samples and are reported in Fig. 1 as the kinetic dispersion estimates.

For both N-fit and O-fit samples, similar trends were observed when a greater number of samples were selected. However, kinetic dispersion reduction was most pronounced for small sample sizes. This can be explained by the decrease in sample fitness when increasing the number of samples. Since IQRNL values were all calculated using the same number of samples, the increasing kinetic dispersion reduction with decreasing number of samples is not expected to be an artifact of biased parameter estimation.

**Scaling of unnecessary duplexes.** The typical unnecessary inter-oligomer duplexes for a given combination of $i$ and $j$ was estimated using the following process. First, 10,000 networks with random

base-sequence were generated. N was calculated individually for each network. Since these networks contain no intentional duplexes, all duplexes are unnecessary duplexes and $N = \Delta N$. The 50th percentile of these N values was taken to be an estimate of the typical unnecessary inter-oligomer duplexes. The 25th and 75th percentile of the N values were taken as lower and upper bounds for this estimate. The unnecessary intra-oligomer duplexes were estimated similarly, except using the O fitness score.

**Generation of new networks.** New networks of oligomers forming less substantial unnecessary duplexes were generated by using *in-silico* optimization of $\Delta N$, $\Delta O$, or $\Delta W_x$. In terms of the design paradigms established by Dirks et al.[84], this generation method contains a network design acting as a positive design component and base-sequence optimization acting as a negative design component.

New networks were generated using the following process described visually in Fig. 3a. First, the intentional duplexes for the network were formalized as a domain-based design[57]. This involved describing each oligomer as a sequence of binding domains and binding domain complements. Binding domains were declared as either fixed (i.e., not to be mutated) or variable (i.e., free to be mutated). Next, initial base-sequences for each domain were specified. The domain-based design and initial domain sequences were used as input for the custom-written Sequence Evolver (abbreviated SeqEvo) computer program, which optimized $\Delta N$, $\Delta O$, or $\Delta W_x$ via a custom evolution-inspired algorithm. Supplementary note 2 details operation of SeqEvo. Supplementary note 3 details fitness score calculation. All mutations performed by SeqEvo rearrange bases within a variable domain. Thus, all oligomers generated by SeqEvo have domains with the same length and base-composition as the initial base-sequences. For the first optimization, default SeqEvo parameters were used. The network resulting from SeqEvo optimization, referred to as a candidate network, was then analyzed by a human looking for obvious flaws. For this purpose, the custom-written Device Profiler (abbreviated DevPro) computer program was used to identify and profile unnecessary duplexes. It is possible to incorporate other software such as the nucleic acid analysis package (NUPACK)[75] thermodynamic simulator at this step if necessary. Based on the human analysis, the candidate network was either accepted as final or the SeqEvo parameters were updated and optimization repeated. The source code of both the SeqEvo and DevPro programs have been made freely available online[55].

**Measurement of in-vitro rate-constants for new networks.** The network design for this model system (Fig. 4a) included three DNA oligomers (labeled $S_1$, $S_2$, and $S_3$) composed of two binding domains (α and β). From their 5' ends: oligomer $S_1$ contains domains α then β, $S_2$ contains the binding complement of β then the binding complement of α, and $S_3$ contains only domain α. These three oligomers form two intentional duplexes: $D_1$ (which forms between $S_1$ and $S_2$) and $D_2$ (which forms between $S_3$ and $S_2$). Domain α is a variable sequence of 10 C's, 10 G's, 10 A's, and 11 T's. Domain β is the fixed sequence TCTCCATG, which was adopted from a previous study in order to mitigate the known effects of toehold stability on hybridization kinetics[85].

Hybridization reactions were characterized using the following procedure. Oligomers were purchased HPLC purified from Integrated DNA Technologies with Cy3 and "Black Hole Quencher 1" modifications, respectively. Oligomer concentrations were calculated using the absorbance coefficients reported by Integrated DNA Technologies and absorption measurements at 260 nm (Thermo Nanodrop One spectrophotometer). Reactants were

prepared in 1x TE buffer (supplemented with 1 M NaCl) at 1 μM concentration. Reactants were combined at an initial concentration of 10 nM in 1x TE (supplemented with 1 M NaCl) inside of one of two Cary Eclipse spectrophotometers. The concentration of unquenched Cy3 dye, which was proportional to the concentration of unreacted $S_1$, was monitored using excitation and emission wavelengths of 548 nanometers and 574 nanometers.

Under these conditions, a plot of inverse-reactant-concentration as a function of time is expected to be linear with slope equal to the reaction rate[22,37,44]. Linear fits were applied to the first 5, 10, 20, 50, 100, 200, and 500 s of data and the fit with the largest coefficient of determination ($R^2$) was taken to be the best rate-constant measurement. This method yielded median $R^2$ values of 0.9968 and 0.9951 for the duplex-formation and oligo-displacement reactions. Supplementary note 4 includes the fluorescence traces, concentration plots, and rate-constant values for each measurement.

The precision of the rate-constant measurements was sampled using triplicate measurements of the "$W_1$-fit-3" oligomer set. This included three measurements each for the formation and displacement reactions at 20 °C and 40 °C. In total, this yielded twelve rate-constant values organized into four groups of three. Within these groups of three, the largest deviation from a group average was 4% and the median deviation was 2%. Errors in commercial oligonucleotide synthesis have been observed to induce up to a factor of two difference in rate-constant values[12] and each group of three rate-constant values were based on a single synthesis. This suggests that, despite the relatively high precision of the experimental method, all measurements for any given oligomer-set may be systematically offset by up to a factor of two due to synthesis errors. Since the observed rate-constant values span four orders of magnitude, it is reasonable to expect the majority of observed rate variation to arise from factors other than synthesis errors. This assumption is further supported by the high reproducibility of certain design groups, such as the $W_1$-fit oligomers at high experimental temperatures.

For Fig. 4d, kinetic dispersion was estimated using the following process. First, the three networks were resampled with replacement 5000 times. For each resampling, the IQRNL was calculated at each of the six temperatures. For each resampling, the median IQRNL of the six temperatures was calculated. The 50th percentile of the median IQRNL across the 6 temperatures was interpreted as the estimated kinetic dispersion. The 25th and 75th percentiles were used as upper and lower bounds for this estimate. Normalized kinetic dispersions were calculated relative to the 50th percentile of the RND group.

**Generation of largest possible networks**. New networks were generated using the following freely available design tools: (1) The SeqEvo computer program, (2) the Domain Design (DD) program[57], (3) the DNASequenceGenerator (DSG) program[54], (4) the Exhaustive Generation of Nucleic Acid Sequence (EGNAS) program[53], (5) the Nucleic Acid Package (NUPACK) software[75], and (6) the Uniquimer3D program[74]. Networks were generated using default program parameters, with as minimal adjustments as possible made to program settings. For reference, networks were also generated by randomly assigning base-sequences. Networks were first generated for a network forming a single 8 base-pair duplex. Either the number of duplexes or the number of base-pairs were repeatedly doubled until 3 runs of the generation program could no longer yield a network satisfying the"no 3's and no 8's" design rule. At this point, the previous network size was revisited. Either the number of duplexes or the number of base-pairs were then increased in 10% increments

until 3 runs of the generation program could no longer yield a network satisfying the "no 3's and no 8's" design rule.

**Reporting summary**. Further information on research design is available in the Nature Portfolio Reporting Summary linked to this article.

## Data availability

Numerical data underlying graphs and charts is provided in the 'Supplementary Data 1' file.

## Code availability

The results reported in this manuscript were either created or duplicated using version 2.0 of the SeqEvo and DevPro programs. The source code of these programs is available on GitHub[55]. The 2.0 versions of these programs have been archived[86] and are also provided in the 'Supplementary Software 1' file.

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

## Acknowledgements

We thank the Nanoscale Materials & Device Research Group at Boise State University. We also thank Dr. Tim Andersen, Dr. James Alexander Liddle, and Dr. Igor Medintz.

We acknowledge the support of the R2 compute cluster (DOI: 10.18122/B2S41H) provided by Boise State University's Research Computing Department. This research was funded by the: (1) W.M. Keck Foundation; (2) Higher Education Research Council (HERC) Idaho Global Entrepreneurial Mission (IGEM), (3) Semiconductor Research Corporation [2018-SB-2842]; (4) National Science Foundation [CMMI 1344915, ECCS 1807809, and ECCS 2227626]; and (5) National Institutes of Health from the National Institute of General Medical Sciences [K25GM093233].

## Author contributions

M.T. performed the research and wrote the manuscript. B.Y. and W.L.H. supervised the research and guided manuscript revisions.

## Competing interests

M.T. founded C-TAG Corporation while this research was in peer-review. The authors declare there are no other competing interests.
