## [Peer Review File · Communications Chemistry]

Reviewers' comments:

Reviewer #1 (Remarks to the Author):

“Generation of DNA Oligomers with Similar Chemical Kinetics via In-Silico Optimization” describes an approach for designing sets of interacting nucleic acid strands with similar hybridization rates. existing datasets for rates of oligo hybridization, creates a heuristic for minimizing kinetic dispersion from dataset mining, develops software to design sets of oligos de novo using the heuristic, and performs a limited amount of experimental validation of the resulting designs. I found the manuscript on the whole to be rigorous and thorough, although at times the dense writing made aspects of the manuscript difficult to follow. My major and minor points are below:

Major points:

1. I struggled at times to understand if the main focus of the manuscript was on the analysis/heuristic or the software. If the former, the discussion of alternative heuristics and the limitations of the chosen heuristic should be expanded. If the latter, the software (which is largely relegated to the supplement) should be afforded a more in-depth description, including its key features/limitations and usage requirements.
2. The statistical analysis of the data presented in the manuscript is lacking. Choices of sample size used for data analysis are not explicitly justified, and a power analysis is not performed. Statistical tests are not performed. The inclusion of such analyses and calculations would help strengthen the conclusions of the manuscript.
3. While the inclusion of experimental data is a strength of the study, it is not clear that the reduction in kinetic dispersion provided by the optimized sets of oligos designed with the software actually improves the outcomes of the experiments beyond reducing kinetic dispersion. While additional experiments may be beyond the scope of the study, the authors should at least provide a more thorough discussion of the broader impacts of the observed results.

Minor points:

1. Some of the terminology used in the manuscript (e.g., "N", "O", "W", "Fitness score") is confusing, as it is either vague or has multiple meanings across different fields. The authors should consider using more specific terminology to improve clarity and minimize potential confusion for readers.

Reviewer #2 (Remarks to the Author):

In this paper, the authors have demonstrated that by optimizing the sequence of DNA in hybridization reactions to minimize undesired intramolecular and intermolecular pairing, is sufficient for a significant reduction in observed variations in reaction kinetics. Several applications involving DNA hybridization reactions rely on consistent kinetics for effectiveness. So, the tools developed by the authors – SeqEvo and DevPro, would be very useful in designing experiments that employ such reactions. The paper is well laid out and covers different aspects including in vitro experiments to justify their claim. The benefits as well as the limitations of their new method has also been well described. Overall it appears to be a great addition to the toolkit for designing DNA oligomers for experiments.

However, the comparison of SeqEvo with other existing methods for DNA oligomer generation is not convincing enough to highlight novelty/ usefulness of SeqEvo. I think the authors should include a few more test cases to highlight the strengths and limitations of their method over others.

Also from an end user perspective, the situations when SeqEvo can and cannot be applied should be clearly laid out, with all size restrictions and target scores for good design. Also since it seems like parameter finetuning is essential for every new network design, some instructions/ tips on how to go about it should also be included perhaps in the supplementary material.

- While the background and motivation for the project is laid out well in the first two paragraphs, I think the authors should include a short paragraph towards the end of the introduction providing a summary of the results and outcome of their endeavors and how it is comparable or different from existing computational tools that are available for the design of DNA oligomers for various experiments.
- In page 5, “while no N-fit, O-fit or W-fit network contained such duplexes larger than 2 base pairs” – which is shown in Fig S4-01, but Figure 4b shows ΔO scores > 100 . If the baseline is driving the ΔO value (which I assume it is, in this case), that should be clearly stated to avoid confusion.
- Figure 4d, legend, “using 5000 resamplings drawn with replacement” – from? While the sample is well defined for a similar calculations in Fig 1c, it is not mentioned what the size of the pool of k values is, from which the samples are drawn. Also can the authors also elaborate on why resampling is necessary?

Other minor comments

- Boltzmann constant should be represented with a capital B in the subscript k_B
- Page 10, “typically scale exponential with network size”, should be exponentially
- Page 10, “For hybridization reactions with where a single”, remove with
- Page 13, “both necessary and unnecessary” duplexes?

Reviewer #3 (Remarks to the Author):

The manuscript "Generation of DNA Oligomers with Similar Chemical Kinetics via In-Silico Optimization" address the problem of kinetic dispersion in hybridization reactions. A potential contributor of kinetic dispersion are unnecessary duplexes which can be present in networks of oligomers. The authors claim that mitigating unnecessary duplexes is the key to achieving low kinetic dispersion. This statement was reinforced by examining existing experimental data and performing in-silico optimization to limit unnecessary duplexes.

The comprehensive results and methods sections, as well as the detailed and ample set of supplemental information, show that the authors put a lot of work and effort into the creation of this manuscript. Any finding and statistical analysis, as well as the outcome of the in-silico optimization are therefore comprehensible, even if the reader is not a chemist nor biologist.

The presented approach of targeting the problem of kinetic dispersion and the corresponding in-silico optimization methods are of importance for the scientific community and I see no reason why this paper shouldn't be accepted.

Reviewer #4 (Remarks to the Author):

Review for Generation of DNA Oligomers with Similar Chemical Kinetics via In-Silico Optimization

Feedback for Authors

In this work, the authors proposed a method of reducing kinetic dispersion of DNA hybridization reactions by designing sequences with in-silico optimization to limit undesired binding between different strands. Here, kinetic dispersion refers to the differences in DNA hybridization reaction kinetics of similar reaction conditions caused by the differences in sequences. The authors started with the analysis of previously published datasets with different DNA hybridization reactions and identified that a) the major contributor to kinetic dispersion is the undesired intra-oligomer duplexes and b) the minor contributor to kinetic dispersion is the undesired inter-oligomer duplexes. Based on this finding, the authors developed an algorithm to design and optimize DNA sequences that limits the undesired intra-oligomer duplexes and inter-oligomer duplexes. The authors showed that adopting such a method may reduce the kinetic dispersion up to 86% by validating with in-vitro kinetic experiments.

The paper would be of interest to many groups within the community and is recommended to be published with the following comments addressed:

1. In page 3, the authors mentioned: "The smallest of these differences is equivalent to approximately 3 unnecessary intra-oligomer duplexes containing 3 base-pairs each." The authors are suggested to

provide a more detailed explanation referring to how to derive the 3 base pair conclusion either in the main text or SI, so the readers may have a better understanding of how this is calculated.

2. Can the authors also provide the histograms of the rate constants post N-fit and O-fit optimization for figure 1b and c in the SI to provide more information regarding where the rate constants converge toward and whether different fitting/optimization lead to different convergence behavior/direction?

3. In figure 1d, the authors presented the optimization results by varying the weight $W(x)$ and showed that kinetic dispersion can be greatly reduced when the weight is heavily biased against limiting undesired intra-oligomer duplexes with $x = 10,000$. However, in the final design when designing sequences for in-vitro validation experiments (figure 4c and 4d), the authors are optimizing the sequences using equal weight, setting $x = 1$. Can the authors explain why and how the weight is chosen for this situation, and how would the authors suggest future readers and users in finding optimal weight values if they choose to adopt the same methods.

4. Finally, would adopting such method greatly reduce and limit the potential design space (and required more computational power) when designing DNA sequences compared to other methods, as it might be more and more difficult to find and design sequences that strictly inhibits even 2 or 3 base-pair of unnecessary intra-oligomer duplexes when people intend to design a large number of interacting sequences, and if so, what would be the suggestion in these cases? (perhaps making the restrictions less strict when designing large library of sequences)

GENERAL RESPONSE TO REFEREE REMARKS

The authors appreciate the constructive feedback from the reviewers. In response to the concerns of the reviewers, the following major changes have been made to the manuscript. (1) The “no 3’s and no 8’s” design rule has been added to the manuscript. This design rule is intended to provide a clearly defined objective for newly generated networks and simplify the discussion surrounding the limitations and advantages of the new generation method. (2) A new subsection titled “Generation of optimized networks using the SeqEvo and DevPro software” has been added to the results section. This new text is intended to clarify the proposed network generation method, describe the software in greater detail, and provide advice for future researchers looking to utilize either the method or software. (3) The comparison to other generation methods has been reworked. This includes replacing the former table 2 with the new figure 6 and replacing the results subsection formerly titled “Comparison to other generation methods” with a new subsection titled “Generation of largest possible networks”. The new subsection includes two new test-cases and is intended to both provide a more detailed comparison with other existing methods and to highlight the advantages of the new method. (4) The discussion has been rewritten. The new discussion is intended to clarify the key results of the study and to communicate more clearly their potential broader impacts. Some text which was previously in the discussion section has been relocated elsewhere in the manuscript.

In addition to these major revisions, the following lesser revisions were also made to the manuscript. (1) A paragraph has been added at the end of the introduction section which surveys the key results of the study. (2) A more thorough statistical analysis has been performed. The text describing this analysis can be found in the “Analysis of kinetic dispersion in existing experimental data” subsection. The added supplemental figures S1-02 to S1-11 provide data supporting this additional statistical analysis. (3) Numerous minor edits have been made to improve the readability of the manuscript, including rewording and/or rearrangement of previously confusing text.

In green text below, we have listed how the modifications to the manuscript have addressed each reviewer’s specific concerns.

RESPONSE TO REVIEWER #1 REMARKS

“Generation of DNA Oligomers with Similar Chemical Kinetics via In-Silico Optimization” describes an approach for designing sets of interacting nucleic acid strands with similar hybridization rates. existing datasets for rates of oligo hybridization, creates a heuristic for minimizing kinetic dispersion from dataset mining, develops software to design sets of oligos de novo using the heuristic, and performs a limited amount of experimental validation of the resulting designs. I found the manuscript on the whole to be rigorous and thorough, although at times the dense writing made aspects of the manuscript difficult to follow. My major and minor points are below:

MAJOR POINTS

1. I struggled at times to understand if the main focus of the manuscript was on the analysis/heuristic or the software. If the former, the discussion of alternative heuristics and the limitations of the chosen heuristic should be expanded. If the latter, the software (which is largely relegated to the supplement) should be afforded a more in-depth description, including its key features/limitations and usage requirements.

The focus of the manuscript is intended to be the relationship between unnecessary duplexes and the large kinetic dispersions sometimes exhibited by networks of interacting DNA oligomers. Several changes have been made to clarify this, including an additional paragraph at the end of the introduction and a rewriting of the discussion section. This comment has been further addressed by the inclusion of a new results subsection titled “Generation of optimized networks using the SeqEvo and DevPro software”, which provides a more in-depth description of the software.

2. The statistical analysis of the data presented in the manuscript is lacking. Choices of sample size used for data analysis are not explicitly justified, and a power analysis is not performed. Statistical tests are not performed. The inclusion of such analyses and calculations would help strengthen the conclusions of the manuscript.

Additional statistical analyses have been added to the manuscript. These can be found in the first results subsection titled “Analysis of kinetic dispersion in existing experimental data”. These additions include the use of a Kolmogorov Smirnov test to compare the rate-constants of the most-fit samples to other samples for the existing experimental data. These additions also include justification for the sample sizes used and the addition of supplemental figures S1-02 to S1-11, which report key statistics calculated for select N-fit and O-fit populations.

3. While the inclusion of experimental data is a strength of the study, it is not clear that the reduction in kinetic dispersion provided by the optimized sets of oligos designed with the software actually improves the outcomes of the experiments beyond reducing kinetic dispersion. While additional experiments may be beyond the scope of the study, the authors should at least provide a more thorough discussion of the broader impacts of the observed results.

The authors believe that additional experiments are beyond the scope of this study. The discussion section of the manuscript has been rewritten and, to address this concern, now includes a more thorough discussion of the broader impacts of the observed results.

MINOR POINTS

1. Some of the terminology used in the manuscript (e.g., "N", "O", "W", "Fitness score") is confusing, as it is either vague or has multiple meanings across different fields. The authors should consider using more specific terminology to improve clarity and minimize potential confusion for readers.

The authors agree that the present language hinders the readability of the paper and can be confusing. Unfortunately, the authors are not aware of any widely acceptable alternatives. To help improve clarity and minimize potential confusion, the text introducing the N, O, and W fitness scores on page 3 has been edited to improve readability.

RESPONSE TO REVIEWER #2 REMARKS

In this paper, the authors have demonstrated that by optimizing the sequence of DNA in hybridization reactions to minimize undesired intramolecular and intermolecular pairing, is sufficient for a significant reduction in observed variations in reaction kinetics. Several applications involving DNA hybridization reactions rely on consistent kinetics for effectiveness. So, the tools developed by the authors – SeqEvo and DevPro, would be very useful in designing experiments that employ such reactions. The paper is well laid out and covers different aspects including in vitro experiments to justify their claim. The benefits as well as the limitations of their new method has also been well described. Overall it appears to be a great addition to the toolkit for designing DNA oligomers for experiments. However, the comparison of SeqEvo with other existing methods for DNA oligomer generation is not convincing enough to highlight novelty/ usefulness of SeqEvo. I think the authors should include a few more test cases to highlight the strengths and limitations of their method over others. Also from an end user perspective, the situations when SeqEvo can and cannot be applied should be clearly laid out, with all size restrictions and target scores for good design. Also since it seems like parameter finetuning is essential for every new network design, some instructions/ tips on how to go about it should also be included perhaps in the supplementary material.

The comparison of SeqEvo to existing methods has been reworked and is now in a results subsection titled “Generation of largest possible networks”. This new section includes two new test cases; (1) the largest possible network of 8-base-pair duplexes and (2) the largest possible single duplex satisfying the “no 3’s and no 8’s” design rule. These results are expected to more convincingly demonstrate the advantages of the SeqEvo software. Additionally, a new subsection titled “Generation of optimized networks using the SeqEvo and DevPro software” has been added to the results section. This subsection contains a description of the situations where SeqEvo can and cannot be applied, as well as tips for tuning program parameters.

- While the background and motivation for the project is laid out well in the first two paragraphs, I think the authors should include a short paragraph towards the end of the introduction providing a summary of the results and outcome of their endeavors and how it is comparable or different from existing computational tools that are available for the design of DNA oligomers for various experiments.

A short paragraph has been added to the end of the introduction section which briefly summarizes key results and outcomes of the study.

- In page 5, “while no N-fit, O-fit or W-fit network contained such duplexes larger than 2 base pairs” – which is shown in Fig S4-01, but Figure 4b shows ΔO scores > 100 . If the baseline is driving the ΔO value (which I assume it is, in this case), that should be clearly stated to avoid confusion.

The text which is referenced by this remark is now on page 9. The paragraph containing this text has been edited to be clearer for readers and now starts “The unnecessary duplexes for each network were characterized using the DevPro program”. The ΔO scores > 100 correctly noted by this reviewer result from the numerous smaller unnecessary intra-oligomer duplexes containing 2 base-pairs or less. Text has been added clarifying this calculation.

- Figure 4d, legend, “using 5000 resamplings drawn with replacement” – from? While the sample is well defined for a similar calculations in Fig 1c, it is not mentioned what the size of the pool of k values is, from which the samples are drawn. Also can the authors also elaborate on why resampling is necessary?

Figure 4’s figure caption has been edited to be less confusing. Text clarifying the process for calculating these kinetic dispersion estimates has been added to the methods subsection titled “Measurement of in-vitro rate-constants for new networks”, which starts on page 20.

OTHER MINOR COMMENTS

- Boltzmann constant should be represented with a capital B in the subscript kB
- Page 10, “typically scale exponential with network size”, should be exponentially
- Page 10, “For hybridization reactions with where a single”, remove with
- Page 13, “both necessary and unnecessary” duplexes?

Edits have been made to correct these issues.

RESPONSE TO REVIEWER #3 REMARKS

The manuscript "Generation of DNA Oligomers with Similar Chemical Kinetics via In-Silico Optimization" address the problem of kinetic dispersion in hybridization reactions. A potential contributor of kinetic dispersion are unnecessary duplexes which can be present in networks of oligomers. The authors claim that mitigating unnecessary duplexes is the key to achieving low kinetic dispersion. This statement was reinforced by examining existing experimental data and performing in-silico optimization to limit unnecessary duplexes.

The comprehensive results and methods sections, as well as the detailed and ample set of supplemental information, show that the authors put a lot of work and effort into the creation of this manuscript. Any finding and statistical analysis, as well as the outcome of the in-silico optimization are therefore comprehensible, even if the reader is not a chemist nor biologist.

The presented approach of targeting the problem of kinetic dispersion and the corresponding in-silico optimization methods are of importance for the scientific community and I see no reason why this paper shouldn't be accepted.

We appreciate the comments of this reviewer and have attempted to maintain the comprehensibility of the manuscript while addressing the concerns of the other reviewers.

RESPONSE TO REVIEWER #4 REMARKS

In this work, the authors proposed a method of reducing kinetic dispersion of DNA hybridization reactions by designing sequences with in-silico optimization to limit undesired binding between different strands. Here, kinetic dispersion refers to the differences in DNA hybridization reaction kinetics of similar reaction conditions caused by the differences in sequences. The authors started with the analysis of previously published datasets with different DNA hybridization reactions and identified that a) the major contributor to kinetic dispersion is the undesired intra-oligomer duplexes and b) the minor contributor to kinetic dispersion is the undesired inter-oligomer duplexes. Based on this finding, the authors developed an algorithm to design and optimize DNA sequences that limits the undesired intra-oligomer duplexes and inter-oligomer duplexes. The authors showed that adopting such a method may reduce the kinetic dispersion up to 86% by validating with in-vitro kinetic experiments. The paper would be of interest to many groups within the community and is recommended to be published with the following comments addressed:

1. In page 3, the authors mentioned: “The smallest of these differences is equivalent to approximately 3 unnecessary intra-oligomer duplexes containing 3 base-pairs each.” The authors are suggested to provide a more detailed explanation referring to how to derive the 3 base pair conclusion either in the main text or SI, so the readers may have a better understanding of how this is calculated.

The text mentioned to by this reviewer is now on page 4. The following text has been modified to clarify this calculation: “The smallest of these differences (3.3×10^3) came from the H22F dataset and was associated with an 84% reduction in kinetic dispersion. Since each unnecessary 3-base-pair intra-oligomer duplex contributes 1×10^3 fitness points to ΔO , the value of 3.3×10^3 is equivalent to approximately 3 unnecessary 3-base-pair duplexes.”

2. Can the authors also provide the histograms of the rate constants post N-fit and O-fit optimization for figure 1b and c in the SI to provide more information regarding where the rate constants converge toward and whether different fitting/optimization lead to different convergence behavior/direction?

Supplemental figures S1-02 to S1-1 have been added to the manuscript. These figures include histograms for the O-fit and N-fit populations referenced by this reviewer.

3. In figure 1d, the authors presented the optimization results by varying the weight $W(x)$ and showed that kinetic dispersion can be greatly reduced when the weight is heavily biased against limiting undesired intra-oligomer duplexes with $x = 10,000$. However, in the final design when designing sequences for in-vitro validation experiments (figure 4c and 4d), the authors are optimizing the sequences using equal weight, setting $x = 1$. Can the authors explain why and how the weight is chosen for this situation, and how would the authors suggest future readers and users in finding optimal weight values if they choose to adopt the same methods.

W_1 optimization was chosen for figure 4 because this was the smallest x which eliminated 3 base-pair unnecessary intra-oligomer-duplexes. Text has been added to the “Kinetic dispersion of newly generated networks” section, which starts on page 8, to clarify this. Advice for future researchers looking to tune this scoring weight has been added with the new “Generation of optimized networks using the SeqEvo and DevPro software” subsection, which starts on page 6.

4. Finally, would adopting such method greatly reduce and limit the potential design space (and required more computational power) when designing DNA sequences compared to other methods, as it might be more and more difficult to find and design sequences that strictly inhibits even 2 or 3 base-pair of unnecessary intra-oligomer duplexes when people intend to design a large number of interacting sequences, and if so, what would be the suggestion in these cases? (perhaps making the restrictions less strict when designing large library of sequences)

The reviewer is correct that adopting the proposed network generation method limits the potential design space. The reworked comparison to other generation methods (figure 6) now explores this limit for two model systems. Suggestions have also been added to the new “Generation of optimized networks using the SeqEvo and DevPro software” subsection suggesting parameters one can tune to generate networks for larger designs.

Reviewers' comments:

Reviewer #1 (Remarks to the Author):

The authors have satisfactorily addressed the bulk of my concerns and improved the manuscript with their revisions. I am supportive of publication in Communications Chemistry provided the following point is addressed:

1. The use of a K-S test to compare distributions of what is essentially a central tendency metric (IQRNL) is not well matched to the point the authors are trying to make. It would be more appropriate to either 1) use a non-parametric test of central tendency such as the Wilcoxon Signed Rank Test or 2) (preferred option) randomly sample the existing data a large number of times, fit that resulting sampling to a parametric distribution (ie, create an empirical distribution), then use the empirical distribution as a means of deriving significance values.

Reviewer #2 (Remarks to the Author):

The authors have addressed all my concerns satisfactorily. I recommend the paper for publication.

Reviewer #4 (Remarks to the Author):

The authors have addressed the comments and suggestions. The introduction of “no 3’s and no 8’s” design rule is a great addition to the paper, as it provides a clean and clear rule in guiding future sequence designs. The new sub-section - “Generation of largest possible networks”, is also really helpful in understanding the differences regarding the studied design software (SeqEvo) and other popular designing softwares. The paper is in good shape and ready to be published.

The following is a list of reviewer remarks (black text) and the authors' response (green text).

REVIEWER #1 REMARKS

The authors have satisfactorily addressed the bulk of my concerns and improved the manuscript with their revisions. I am supportive of publication in Communications Chemistry provided the following point is addressed:

1. The use of a K-S test to compare distributions of what is essentially a central tendency metric (IQRNL) is not well matched to the point the authors are trying to make. It would be more appropriate to either 1) use a non-parametric test of central tendency such as the Wilcoxon Signed Rank Test or 2) (preferred option) randomly sample the existing data a large number of times, fit that resulting sampling to a parametric distribution (ie, create an empirical distribution), then use the empirical distribution as a means of deriving significance values.

The authors agree that the use of a Kolmogorov-Smirnov (K-S) test to compare distributions of IQRNL values is not appropriate. In the prior version of the manuscript, the K-S test was instead used to compare distributions of rate constant values, which is more appropriate. We apologize for the unclear wording of the prior version of the manuscript and have made edits to clarify this. These edits can be found in the results section under the "Analysis of kinetic dispersion in existing experimental data" sub-heading. All modified text has been colored blue. We have also added p-values calculated from the Wilcoxon Signed Rank Test to the supplemental material. The p-values from the Wilcoxon Signed Rank Test follow a similar trend to the p-values of the K-S test, although the Wilcoxon Signed Rank Test appears to require more samples to reach the same conclusion.

REVIEWERS' COMMENTS:

Reviewer #1 (Remarks to the Author):

The authors have satisfactorily addressed my concerns. I support publication in Communications Chemistry.